# Mechano-regulation of GLP-1 production by Piezo1 in intestinal L cells

Yanling Huang[1†], Haocong Mo[1†], Jie Yang[2†], Luyang Gao[1†], Tian Tao[1], Qing Shu[1], Wenying Guo[1], Yawen Zhao[1], Jingya Lyu[1], Qimeng Wang[3], Jinghui Guo[4], Hening Zhai[5], Linyan Zhu[6], Hui Chen[3*], Geyang Xu[1,7*]

[1]Department of Physiology, School of Medicine, Jinan University, Guangzhou, China; [2]Department of Pathology, School of Basic Medicine, Guangzhou Medical University, Guangdong, China; [3]Biotherapy Center, Cell-gene Therapy Translational Medicine Research Center, The Third Affiliated Hospital of Sun Yat-Sen University, Guangzhou, China; [4]School of Medicine, The Chinese University of Hong Kong, Shenzhen, China; [5]Endoscopy Center, The First Affiliated Hospital of Jinan University, Guangzhou, China; [6]Department of Pharmacology, School of Medicine, Jinan University, Guangzhou, China; [7]Key Laboratory of Viral Pathogenesis & Infection Prevention and Control (Jinan University), Ministry of Education, Guangzhou, China

**\*For correspondence:**
chenh567@mail.sysu.edu.cn (HC);
xugeyangliang@163.com (GX)

†These authors contributed equally to this work

**Competing interest:** The authors declare that no competing interests exist.

## eLife Assessment

This study focuses on the regulation of GLP-1 in enteroendocrine L cells and how this may be stimulated by the mechanogated ion channel Piezo1 and the CaMKKbeta-CaMKIV-mTORC1 signaling pathway. The work is innovative and is considered **valuable**, as the hypothesis that is being tested may have significant mechanistic and translational implications. Data to support the proposed mechanism were considered incomplete, yet data to support the overall physiological characterization were considered **solid**.

**Abstract** Glucagon-like peptide 1 (GLP-1) is a gut-derived hormone secreted by intestinal L cells and vital for postprandial glycemic control. As open-type enteroendocrine cells, whether L cells can sense mechanical stimuli caused by chyme and thus regulate GLP-1 synthesis and secretion is unexplored. Molecular biology techniques revealed the expression of Piezo1 in intestinal L cells. Its level varied in different energy status and correlates with blood glucose and GLP-1 levels. Mice with L cell-specific loss of Piezo1 (*Piezo1* IntL-CKO) exhibited impaired glucose tolerance, increased body weight, reduced GLP-1 production and decreased CaMKKβ/CaMKIV-mTORC1 signaling pathway under normal chow diet or high-fat diet. Activation of the intestinal Piezo1 by its agonist Yoda1 or intestinal bead implantation increased the synthesis and secretion of GLP-1, thus alleviated glucose intolerance in diet-induced-diabetic mice. Overexpression of Piezo1, Yoda1 treatment or stretching stimulated GLP-1 production and CaMKKβ/CaMKIV-mTORC1 signaling pathway, which could be abolished by knockdown or blockage of Piezo1 in primary cultured mouse L cells and STC-1 cells. These experimental results suggest a previously unknown regulatory mechanism for GLP-1 production in L cells, which could offer new insights into diabetes treatments.

## Introduction

The gastrointestinal (GI) tract represents the largest endocrine organ in the human body. The enteroendocrine cells (EECs) located throughout the GI tract secrete a large number of gastrointestinal

hormones to regulate a variety of physiological processes and are key regulators for energy homeostasis (*Bany Bakar et al., 2023*). GLP-1 is one of the gut-derived peptide hormones essential for postprandial glycemic control (*Song et al., 2019*). It is produced from Proglucagon (Gcg) by proprotein convertase in the intestinal L cells, a group EECs predominantly situated in the distal gut (*Drucker, 2006*; *Rouillé et al., 1997*). The circulating GLP-1 levels rapidly increase after meal and reduce postprandial blood glucose fluctuations by augmenting insulin secretion, suppressing glucagon secretion and slowing gastric emptying (*Drucker, 2006*; *Willms et al., 1996*). Nowadays, GLP-1-based therapy is well-recognized and commonly used in treatment of type 2 diabetes mellitus (T2DM; *Saxena et al., 2021*; *Tan et al., 2022*). Elucidation of the mechanism that regulates GLP-1 production is essential for the development of new drug targets for the treatment of diabetes.

EECs can be divided into two categories according to their morphology: open type and closed type. The open type EECs possess microvilli protruding into the gut lumen and have direct contact with the luminal contents. In contrast, the closed type EECs are located basolaterally without direct contact with the lumen (*Gribble and Reimann, 2016*). Both types of EECs synthesize and store peptides or hormones in secretory granules and release them by exocytosis at the basolateral membrane (*Atanga et al., 2023*). As open-type EECs, L cells received both chemical and mechanical signals from the luminal contents, and neural signals from the nerves (*Furness et al., 2013*). It has been well-documented that nutrients such as glucose, lipids, and amino acids in the intestinal lumen can stimulate the secretion of GLP-1 from L cells (*Diakogiannaki et al., 2012*). GLP-1 secretion can also be stimulated by intrinsic cholinergic nerves (*Anini et al., 2002*; *Drucker, 2006*). However, whether and how L cells coordinate mechanical stimuli from intestinal lumen to regulate GLP-1 production remain poorly understood.

Piezo channels, including Piezo1 and Piezo2 have recently been identified as mechanosensitive ion channels involved in the sensation of multiple mechanical stimuli, such as shear stress, pressure, and stretch (*Gudipaty et al., 2017*; *Li et al., 2014*; *Romac et al., 2018*). They allow the influx of cations such as $Ca^{2+}$ and $Na^+$ in response to mechanical tension and converts mechanical stimuli into various electrical and chemical signals. Piezo1 plays a crucial role in blood pressure regulation, red blood cell volume regulation, bone homeostasis, pulmonary and cardiac functions (*Cahalan et al., 2015*; *Lai et al., 2022*; *Wang et al., 2023*; *Wang et al., 2016*). Previous studies have reported that Piezo1 is expressed in the intestinal epithelium, regulating gut peristalsis, barrier function, mucus secretion, and inflammation (*Jiang et al., 2021*; *Liu et al., 2022a*; *Sugisawa et al., 2020*; *Xu et al., 2021*). Interestingly, accumulating evidence demonstrates the regulation of insulin and ghrelin secretion by Piezo1 (*Deivasikamani et al., 2019*; *Ye et al., 2022*; *Zhao et al., 2024*). Recent studies have also reported that Piezo2 is expressed in a population of EECs and convert force into serotonin release (*Alcaino et al., 2018*; *Treichel et al., 2022*). These findings suggest a critical role of Piezo channels in the mechano-regulation of hormone production. However, whether Piezo channels are expressed L cells and play a role in GLP-1 production remain unknown.

The current study has shown that Piezo1 channels on intestinal L cells mediate mechanosensing of intestinal contents and regulate glucose homeostasis by triggering GLP-1 synthesis and secretion via the CaMKKβ/CaMKIV-mTORC1 signaling pathway. This finding provides new insights into the treatment of T2DM and lays a theoretical foundation for the development of antidiabetic drugs targeting Piezo1.

## Results

### Assessment of Piezo1 in human and mouse intestine in different energy status

*Piezo1* mRNA was found to be highly expressed in both mouse ileal mucosa and STC-1 cells (*Figure 1—figure supplement 1A*). Moreover, Piezo1 was co-localized with GLP-1 in immunofluorescent staining on NCD fed mouse ileal sections, indicating its expression in L cells (*Figure 1—figure supplement 1B*). Interestingly, increased body weight and impaired glucose tolerance were observed in high-fat diet-induced diabetic mice, while Piezo1 and Proglucagon expression levels in the ileal mucosa of diabetic mice were significantly lower than that in mice feed with normal chow diet (*Figure 1—figure supplement 1C–F*). Moreover, ileal mucosal *Piezo1* mRNA levels were positively correlated with *Gcg* mRNA levels (*Figure 1—figure supplement 1G*), but negatively correlated with the AUC of glucose

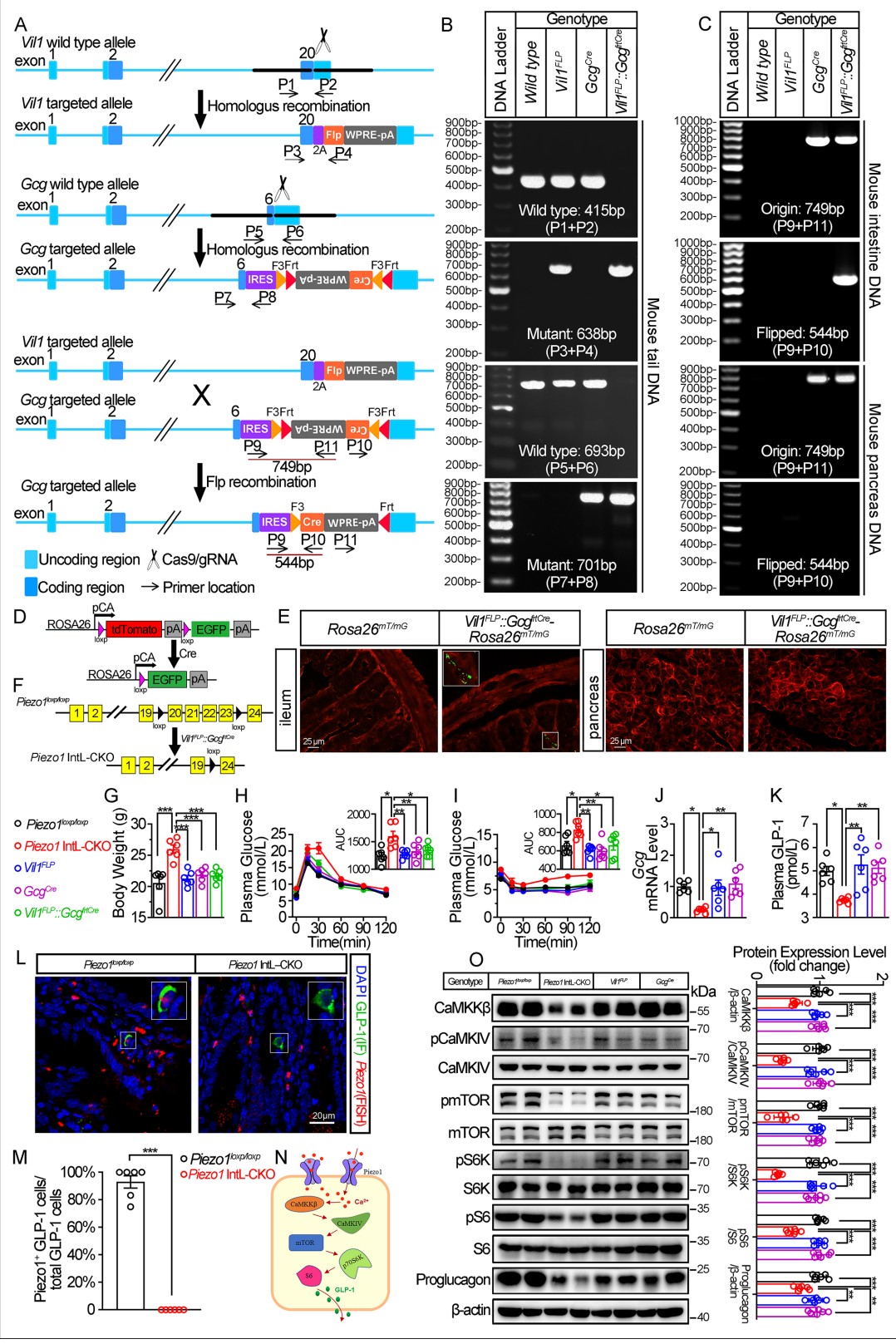

**Figure 1.** Generation, validation, and characterization of *Piezo1* IntL-CKO mice. (**A**) Schematic description for the generation of *Vil1^FLP^* and Flippase-dependent *Gcg^Cre^* mice. *Vil1^FLP^* flip the inverted Cre gene in the *Gcg^Cre^* cassette in *Vil1^FLP^::Gcg^frtCre^* mice to restrict Cre expression in intestinal L cells. As shown, locations of genotyping primers are also indicated. (**B**) Tail DNA genotyping PCR results using genotyping primer for *Vil1^FLP^*, *Gcg^Cre^* and Flippase-activated Cre (*Vil1^FLP^::Gcg^frtCre^*) mice. (**C**) Intestine and pancreas DNA genotyping results. The 'Original' band represents the original *Gcg^Cre^* cassette with

*Figure 1 continued on next page*

*Figure 1 continued*

inverted Cre, while the 'Flipped' band represents recombined $Gcg^{Cre}$ cassette with Cre flipped into the correct direction. (**D**) Schematic description for the validation of $Vil1^{FLP}::Gcg^{frtCre}$ efficacy by crossing with $Rosa26^{mT/mG}$ reporter mice. (**E**) Fluorescence was detected in the ileal and pancreatic tissues from $Rosa26^{mT/mG}$ and $Vil1^{FLP}::Gcg^{frtCre}$-$Rosa26^{mT/mG}$ mice by frozen tissue confocal microscopy. Green fluorescence represents successful deletion of TdTomato and reactivation of EGFP in the Cre-expressing cells. (**F**) Schematic description for the generation of Intestinal L cell-$Piezo1^{-/-}$ mice ($Piezo1$ IntL-CKO) by crossing $Piezo1^{loxp/loxp}$ mice with $Vil1^{FLP}::Gcg^{frtCre}$ mice. (**G**) Body weight of 14- to 16-week-old male mice of the indicated genotypes fed with NCD (n=6/group). (**H, I**) IPGTT (**H**) and ITT (**I**) and associated area under the curve (AUC) values of 14- to 16-week-old male mice of the indicated genotypes fed with NCD (n=6/group). (**J**) $Gcg$ mRNA levels in ileum of 14- to 16-week-old male mice of the indicated genotypes fed with NCD (n=6/group). (**K**) The plasma GLP-1 levels in 14- to 16-week-old male mice of the indicated genotypes fed with NCD (n=6/group). (**L**) Representative images for $Piezo1$ RNA-FISH and GLP-1 immunofluorescent staining in the ileum of 14-week-old male mice of indicated genotypes fed with NCD (n=6/group). (**M**) Percentage of Piezo1-positive GLP-1 cells in total GLP-1 cells in the ileal mucosa of 14-week-old male mice of indicated genotypes fed with NCD (n=6/group). (**N**) A schematic diagram depicting the potential mechanisms linking the CaMKKβ/CaMKIV-mTOR signaling pathway and GLP-1 production. (**O**) Representative western blots are shown for indicated antibodies in the ileal mucosa (n=6/group). Data are represented as mean ± SEM. Significance was determined by Student's t test for comparison between two groups, and by one-way ANOVA for comparison among three groups or more, *p<0.05, **p<0.01, ***p<0.001.

The online version of this article includes the following source data and figure supplement(s) for figure 1:

**Source data 1.** PDF file containing original gels and blots for *Figure 1B, C and O*, indicating the relevant bands and treatments.

**Source data 2.** Original files for gel and western blot analysis displayed in *Figure 1B, C and O*.

**Source data 3.** Original data for *Figure 1*.

**Figure supplement 1.** Assessment of Piezo1 and GLP-1 in mouse and human ilea.

**Figure supplement 1—source data 1.** PDF file containing original western blots for *Figure 1—figure supplement 1E*, indicating the relevant bands and treatments.

**Figure supplement 1—source data 2.** Original files for western blot analysis displayed in *Figure 1—figure supplement 1E*.

**Figure supplement 1—source data 3.** Original data for *Figure 1—figure supplement 1*.

**Figure supplement 2.** Food intake and water intake of *Piezo1* IntL-CKO mice.

**Figure supplement 2—source data 1.** Original files for food intake analysis displayed in *Figure 1—figure supplement 2*.

**Figure supplement 3.** *Piezo1* IntL-CKO mice preserve normal pancreatic morphology and Proglucagon expression under normal diet feeding.

**Figure supplement 3—source data 1.** PDF file containing original western blots for *Figure 1—figure supplement 3D*, indicating the relevant bands and treatments.

**Figure supplement 3—source data 2.** Original files for western blot analysis displayed in *Figure 1—figure supplement 3D*.

**Figure supplement 3—source data 3.** Original data for *Figure 1—figure supplement 3*.

**Figure supplement 4.** Intestinal morphology of *Piezo1* IntL-CKO mice.

**Figure supplement 4—source data 1.** Original files for length of small intestine analysis displayed in *Figure 1—figure supplement 4C, D*.

**Figure supplement 5.** Double immunostaining of Piezo1 and GLP-1 in the intestines of *Piezo1* IntL-CKO mice.

**Figure supplement 5—source data 1.** Original files for the analysis of the percentage of Piezo1-positive GLP-1 cells among total GLP-1 cells in various regions of the intestinal mucosa are shown in *Figure 1—figure supplement 5*.

**Figure supplement 6.** Expression of Piezo1 in intestinal ghrelin cells and pancreatic α cells.

**Figure supplement 7.** Assessment of L cell hormones and CCK in the ileum of *Piezo1* IntL-CKO mice.

**Figure supplement 7—source data 1.** Original data for *Figure 1—figure supplement 7B, D and E*.

**Figure supplement 8.** Effect of L cell-specific *Piezo1* deletion on intestinal barrier function and tight junction proteins.

**Figure supplement 8—source data 1.** PDF file containing original western blots for *Figure 1—figure supplement 8C*, indicating the relevant bands and treatments.

**Figure supplement 8—source data 2.** Original files for western blot analysis displayed in *Figure 1—figure supplement 8C*.

---

tolerance test (*Figure 1—figure supplement 1H*). Obese T2DM patients who underwent Roux-en-Y gastric bypass (RYGB) surgery showed decreased BMI (*Figure 1—figure supplement 1I*) and increased Piezo1 and GLP-1 in ileal mucosa (*Figure 1—figure supplement 1J, K*) compared to that before surgery. These findings indicated that Piezo1 is expressed in intestinal L cells and its level varies in different energy status.

## Generation and characterization of *Piezo1* IntL-CKO mice

To investigate the potential role of Piezo1 in GLP-1 production, we tried to knockout *Piezo1* in L cells by Cre-loxP system driven by an L cell-specific promoter. Proglucagon (encoded by *Gcg* gene)

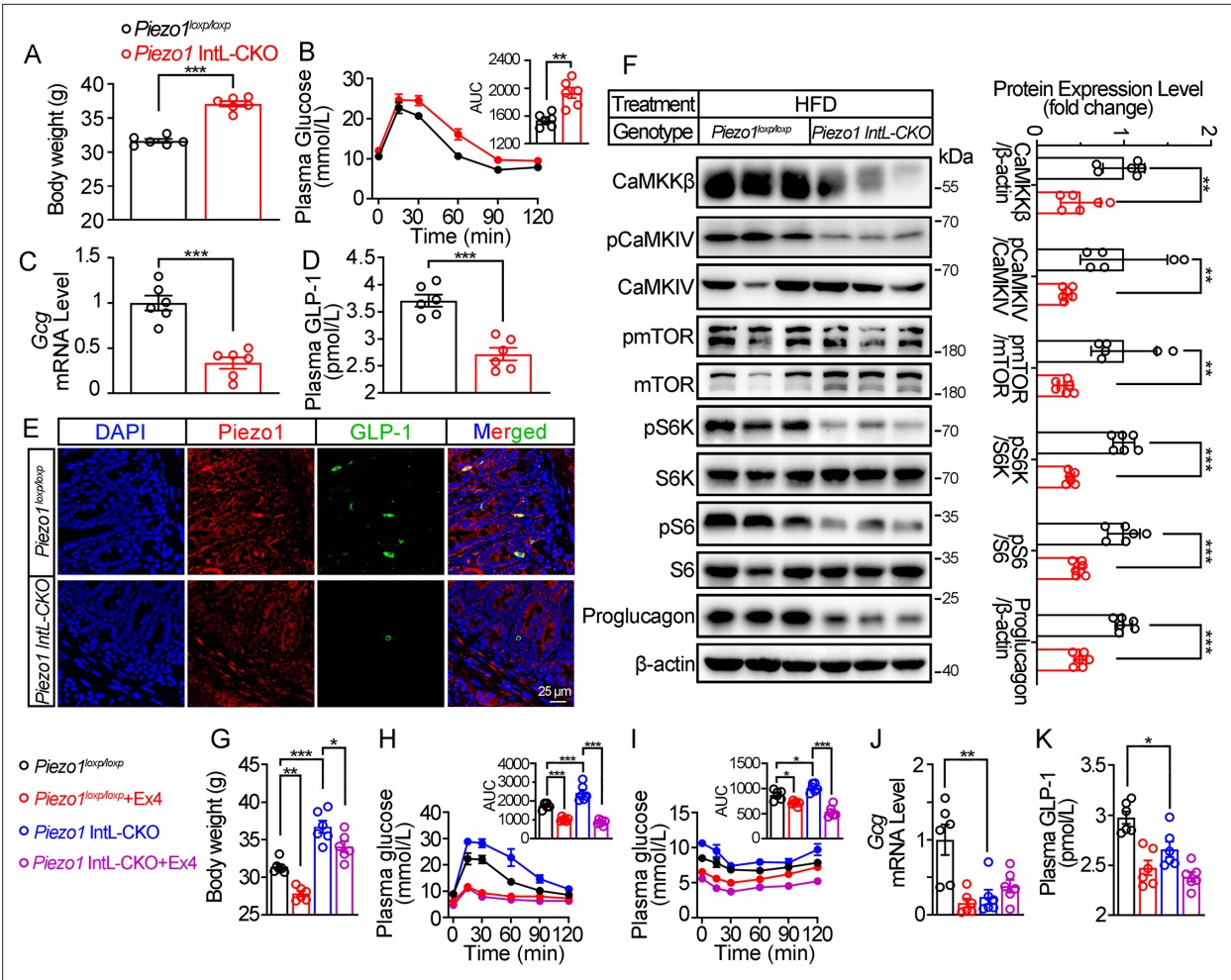

**Figure 2.** Validation and phenotype of *Piezo1* IntL-CKO mice fed with high-fat diet. (**A**) Body weight of 14- to 16-week-old male *Piezo1^loxp/loxp^* and *Piezo1* IntL-CKO mice fed with HFD for 10 weeks (n=6/group). (**B**) IPGTT and associated area under the curve (AUC) values of 14- to 16-week-old male *Piezo1^loxp/loxp^* and *Piezo1* IntL-CKO mice fed with HFD (n=6/group). (**C**) *Gcg* mRNA levels in the ileal mucosa of 14- to 16-week-old male *Piezo1^loxp/loxp^* and *Piezo1* IntL-CKO mice fed with HFD (n=6/group). (**D**) The plasma GLP-1 level in 14- to 16-week-old male *Piezo1^loxp/loxp^* and *Piezo1* IntL-CKO mice fed with HFD (n=6/group). (**E**) Double immunofluorescent staining of Piezo1, and GLP-1 in the ilea of 14- to 16-week-old male *Piezo1^loxp/loxp^* and *Piezo1* IntL-CKO mice fed with HFD (n=6/group). (**F**) Representative western blots are shown for indicated antibodies in the ileal mucosa (n=6/group). (**G**) Body weight after 7 consecutive days infusion of saline or Ex-4 (100 μg/kg body weight) in 14- to 16-week-old male *Piezo1^loxp/loxp^* and *Piezo1* IntL-CKO mice fed with HFD (n=6/group). (**H, I**) IPGTT (**H**) and ITT (**I**) and associated area under the curve (AUC) values after consecutive infusion of saline or Ex-4. (**J**) *Gcg* mRNA levels in the ileal mucosa (n=6/group) after consecutive infusion of saline or Ex-4. (**K**) The plasma GLP-1 level after consecutive infusion of saline or Ex-4 (n=6/group). Data are represented as mean ± SEM. Significance was determined by Student's t test for comparison between two groups, and by one-way ANOVA for comparison among three groups or more, *p<0.05, **p<0.01, ***p<0.001.

The online version of this article includes the following source data and figure supplement(s) for figure 2:

**Source data 1.** PDF file containing original western blots for *Figure 2F*, indicating the relevant bands and treatments.

**Source data 2.** Original files for western blot analysis displayed in *Figure 2F*.

**Source data 3.** Original data for *Figure 2*.

**Figure supplement 1.** *Piezo1* IntL-CKO mice preserve normal pancreatic morphology and proglucagon expression under HFD.

**Figure supplement 1—source data 1.** PDF file containing original western blots for *Figure 2—figure supplement 1D*, indicating the relevant bands and treatments.

**Figure supplement 1—source data 2.** Original files for western blot analysis displayed in *Figure 2—figure supplement 1D*.

**Figure supplement 1—source data 3.** Original data for *Figure 2—figure supplement 1*.

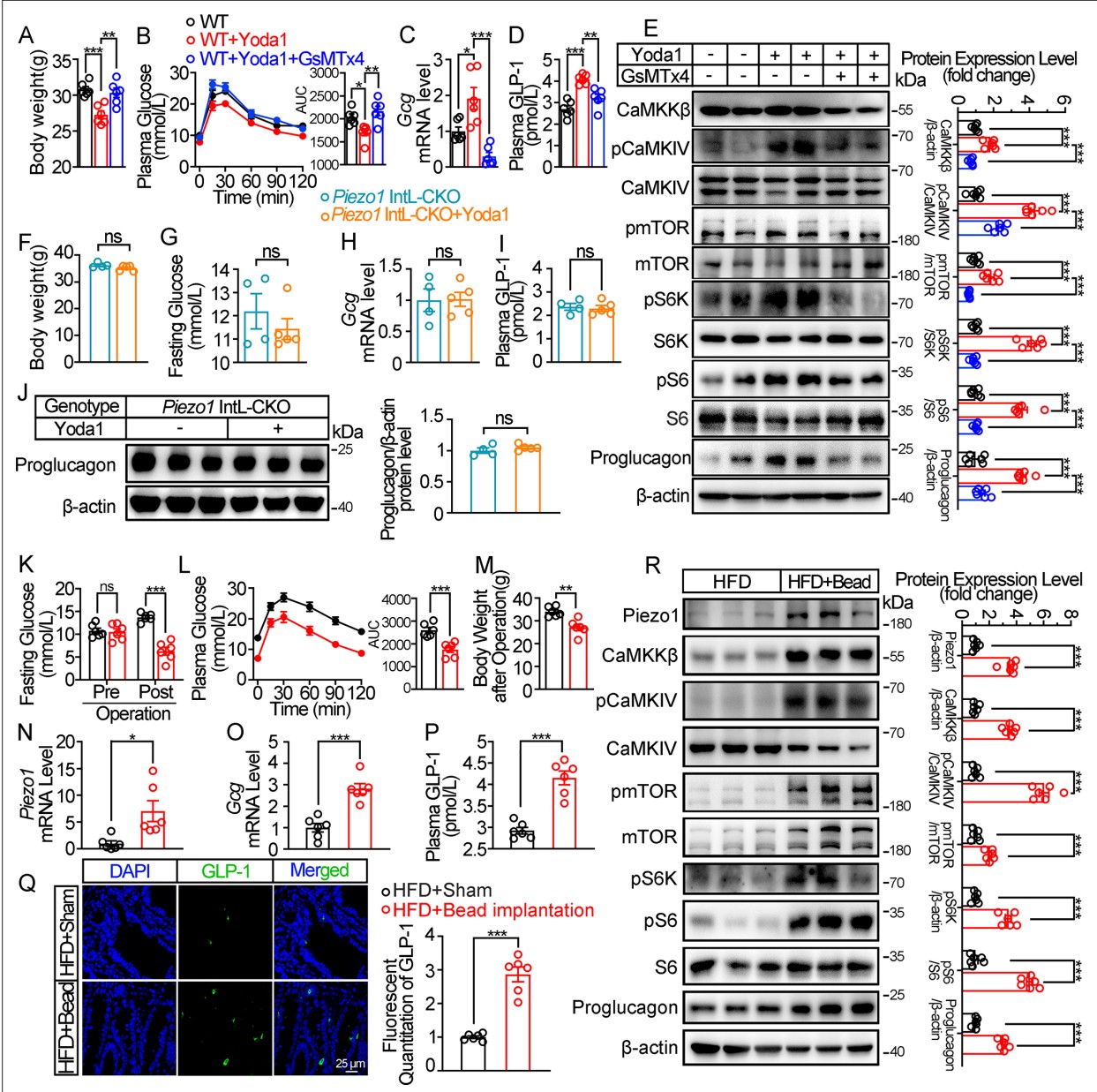

**Figure 3.** Chemical and mechanical interventions of Piezo1 regulate GLP-1 synthesis in mice. (**A–E**) 14- to 16-week-old male C57BL/6 J mice fed with HFD for 10 weeks were infused with vehicle, Yoda1 (2 µg per mouse) or GsMTx4 (250 µg/kg) by i.p. for 7 consecutive days. (n=6/group). (**A**) Body weight after consecutive drug infusion. (**B**) IPGTT and associated area under the curve (AUC) values. (**C**) *Gcg* mRNA levels in the ileal mucosa. (**D**) Plasma GLP-1. (**E**) Representative western blots are shown for indicated antibodies in the ileal mucosa. (**F–J**) 14- to 16-week-old male *Piezo1* IntL-CKO mice fed with HFD for 10 weeks were infused with vehicle, Yoda1 (2 µg per mouse) by i.p. for 7 consecutive days. (n=4 or 5/group). (**F**) Body weight after 7 consecutive days' drug infusion. (**G**) Fasting blood glucose levels. (**H**) Ileal mucosal *Gcg* mRNA levels. (**I**) Plasma GLP-1 levels. (**J**) Ileal mucosal Proglucagon protein levels. (**K–R**) 14- to 16-week-old male C57BL/6 J mice fed with HFD were subjected to sham operation, or intestinal bead implantation (n=6/group). (**K**) Fasting blood glucose levels. (**L**) IPGTT and associated area under the curve (AUC) values. (**M**) Body weight. (**N, O**) *Piezo1* (**N**) and *Gcg* (**O**) mRNA levels in the ileal mucosa. (**P**) The plasma GLP-1 levels. (**Q**) Immunofluorescence staining of GLP-1 in ileum and quantification of GLP-1-positive cells. (**R**) Representative western blots images and densitometry quantification for indicated antibodies in the ileal mucosa. Data are represented as mean ± SEM. Significance was determined by Student's t test for comparison between two groups, and by one-way ANOVA for comparison among three groups or more, *p<0.05, **p<0.01, ***p<0.001.

The online version of this article includes the following source data and figure supplement(s) for figure 3:

**Source data 1.** PDF file containing original western blots for *Figure 3E, J and R*, indicating the relevant bands and treatments.

**Source data 2.** Original files for western blot analysis displayed in *Figure 3E, J and R*.

*Figure 3 continued on next page*

Figure 3 continued

Source data 3. Original data for *Figure 3*.

Figure supplement 1. Effect of intestinal bead implantation on fecal weight, gastrointestinal transit time and abdominal pain in C57BL/6 J mice.

Figure supplement 1—source data 1. Original data for *Figure 3—figure supplement 1*.

is mainly expressed in both L cells and pancreatic α cells (*Jin, 2008*). Villin-1 (encoded by *Vil1* gene) is expressed in gastrointestinal epithelium, including L cells, but not in pancreatic α cells (*Maunoury et al., 1992*; *Rutlin et al., 2020*). Since neither Gcg nor Villin are specific markers for L cells, we tried to generate a new line of mice enabling loss of Piezo1 expression specifically in the intestine L cell by combination of FLP-Frt and Cre-loxP system. We inserted a Flippase (FLP) expression cassette in the 3'UTR of *Vil1* to generate a *Vil1* promoter-driven FLP mice (*Vil1^{FLP}*; *Figure 1A*). Then, we generated Flippase-dependent *Gcg* promoter driven-Cre (*Gcg^{Cre}*) mice by inserting an Frt-flanked Cre expression cassette in reverse orientation within the 3'- UTR of *Gcg* gene (*Figure 1A*). We further crossed the *Vil1^{FLP}* mice with *Gcg^{Cre}* mice to obtain L-cell-specific Cre mice (*Vil1^{FLP}::Gcg^{frtCre}*), in which *Vil1* promoter-driven Flippase flipped the reverse Cre cassette into a correct orientation in Villin-positive cells (including L cells, but not pancreatic α cells), and thus Cre can only be expressed under the *Gcg* promoter in L cells. The genotypes of the *Vil1^{FLP}*, *Gcg^{Cre}* and *Vil1^{FLP}::Gcg^{frtCre}* mice were identified by PCR with specific primers (*Figure 1B*). The flipping of the reverse Cre cassette was validated by PCR, which confirmed that the flipping only occurred the intestine, but not in the pancreas (*Figure 1C*). To confirm the cell type specificity of Cre activity, we crossed *Vil1^{FLP}::Gcg^{frtCre}* mice to *Rosa26^{mT/mG}* reporter mice. All tissues and cells of *Rosa26^{mT/mG}* mice express red fluorescence (membrane-targeted tdTomato; mT) at baseline, and switch to membrane-targeted EGFP in the presence of cell-specific Cre (*Figure 1D*). EGFP expression was only observed scatteredly in the intestine, but not in the pancreas, indicating the intestine-specific Cre activity in the *Vil1^{FLP}::Gcg^{frtCre}* mice (*Figure 1E*). Finally, we bred *Vil1^{FLP}::Gcg^{frtCre}* mice with *Piezo1^{loxp/loxp}* mice to generate *Piezo1* IntL-CKO mice (*Figure 1F*).

Under normal chow diet, *Piezo1* IntL-CKO mice exhibited increased body weight (*Figure 1G*) and greater glycemic excursions compared to control groups (*Piezo1^{loxp/loxp}*, *Vil1^{FLP}*, *Gcg^{Cre}* and *Vil1^{FLP}::Gcg^{frtCre}*; *Figure 1H and I*), while the food and water intake were not changed (*Figure 1—figure supplement 2A, B*). The morphology of islet (*Figure 1—figure supplement 3A*) and ileum (*Figure 1—figure supplement 4A*) were not affected. Ileal mucosal Proglucagon expression and plasma GLP-1 level were significantly lower in *Piezo1* IntL-CKO mice than that in all littermate controls such as *Piezo1^{loxp/loxp}*, *Vil1^{FLP}*, *Gcg^{Cre}* and *Vil1^{FLP}::Gcg^{frtCre}* mice (*Figure 1J and K*), while no significant alteration was observed in the expression of pancreatic Piezo1 and Proglucagon (*Figure 1—figure supplement 3B–D*). According to in situ hybridization of Piezo1 and immunofluorescence analysis of GLP-1, the expression of Piezo1 disappeared in GLP-1 positive cells, suggesting successful knockout of Piezo1 in L cells in *Piezo1* IntL-CKO mice (*Figure 1L and M*). Also depicted in *Figure 1—figure supplement 5*, Piezo1 is expressed in GLP-1-positive cells of the duodenum, jejunum, ileum, and colon in control mice, but not in *Piezo1* IntL-CKO mice. However, Piezo1 remains expressed in intestinal ghrelin positive cells and pancreatic glucagon-positive cells of *Piezo1* IntL-CKO mice (*Figure 1—figure supplement 6*). Moreover, while GLP-1 levels were reduced in L cells of *Piezo1* IntL-CKO mice, levels of PYY, another hormone secreted by L cells, were unaffected (*Figure 1—figure supplement 7A–D*). Additionally, ileal mucosal cholecystokinin (CCK), a hormone secreted by I cells with metabolic effects similar to GLP-1, was also unchanged in *Piezo1* IntL-CKO mice (*Figure 1—figure supplement 7E*). Previous study showed that Piezo1 affected intestinal tight junctions and epithelial integrity (*Jiang et al., 2021*). To access whether loss of Piezo1 in L cells affect epithelial integrity of the intestine, we examined the expression of tight junction proteins, including ZO-1 and Occludin. As shown in *Figure 1—figure supplement 8*, the expression of ZO-1 and Occludin remained unchanged in *Piezo1* IntL-CKO mice when compared to littermate controls.

Piezo1 is a non-selective cationic channel that allows passage of $Ca^{2+}$ and $Na^+$. CaMKKβ is the main calcium/calmodulin dependent protein kinase kinase involved in the regulation of metabolic homeostasis (*Marcelo et al., 2016*). It is activated by binding calcium-calmodulin ($Ca^{2+}$/CaM), resulting in downstream activation of kinases CaMKIV. The activation of CaMKIV modulate the gene expression of nutrient- and hormone-related proteins (*Ban et al., 2000*; *Chen et al., 2011*; *Takemoto-Kimura et al., 2017*). Previous studies have reported that $Ca^{2+}$ and mTOR signaling regulate the production

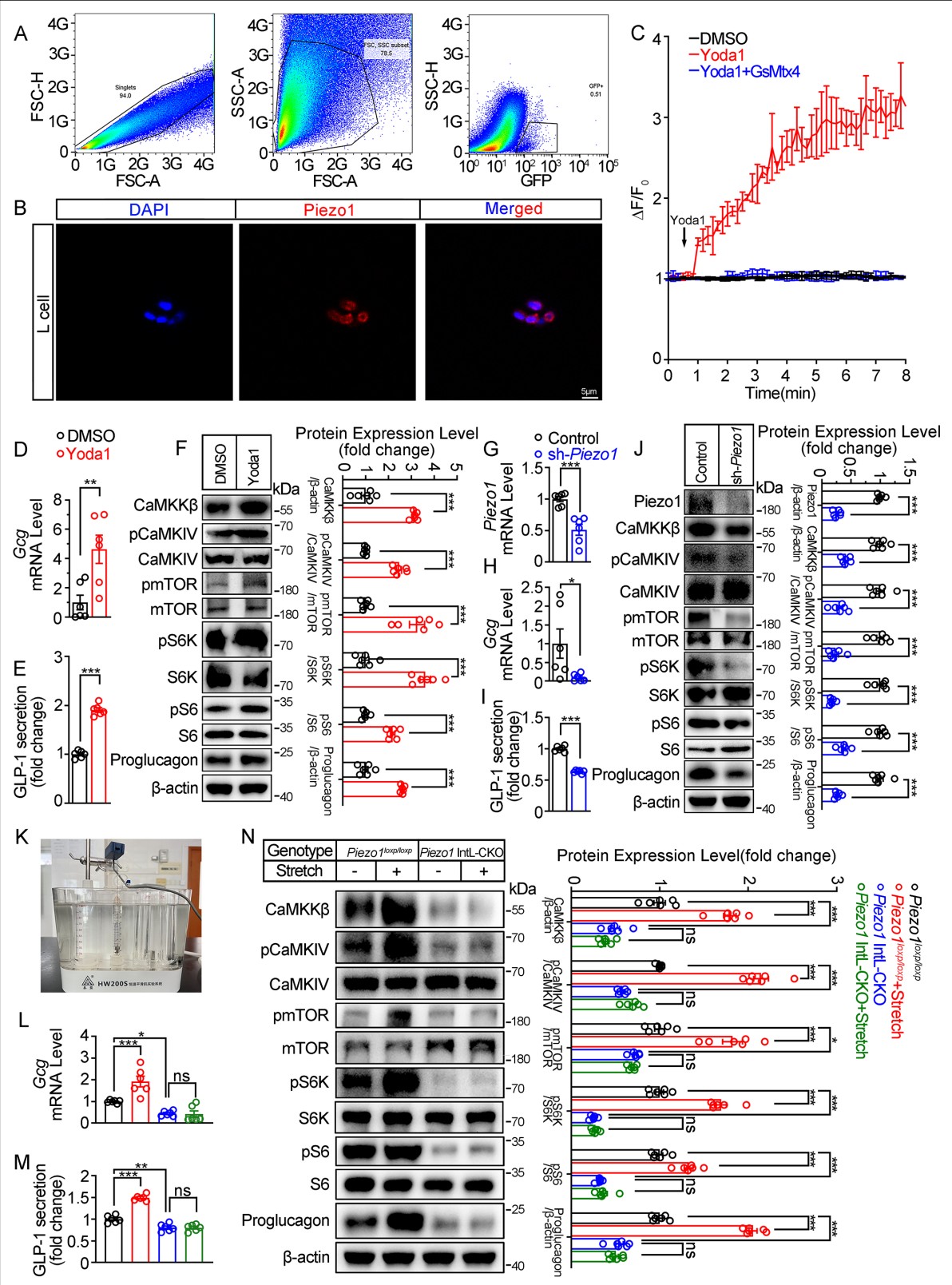

**Figure 4.** Piezo1 regulates GLP-1 synthesis and secretion in primary cultured mouse L cells and isolated mouse ileum. (**A**) Isolation of mouse L cells (GFP positive) from ileal tissue by FACS. The gating in flowcytometry for sorting of GFP-positive cells. (**B**) Immunofluorescent staining of Piezo1 in sorted GFP-positive L cells. (**C**) Intracellular Ca²⁺ imaging by fluo-4-AM calcium probe. The change of fluorescent intensity ($\Delta$F/F0) was plotted against time. (**D–F**) L cells were treated with vehicle or Yoda1 (5 µM) for 24 hr. (**D**) *Gcg* mRNA expression. (**E**) GLP-1 concentrations in the culture medium. (**F**) Western

*Figure 4 continued on next page*

*Figure 4 continued*

blot images and densitometry quantification for the indicated antibodies. (**G–J**) Knockdown of Piezo1 in L cells by shRNA for 48 hours. (**G**) *Piezo1* mRNA expression. (**H**) *Gcg* mRNA expression. (**I**) GLP-1 levels in the culture medium. (**J**) Western blot images and densitometry quantification for the indicated antibodies. (**K–N**) Ileal tissues from *Piezo1*^loxp/loxp^ and *Piezo1* IntL-CKO mice were subjected to tension force (n=6/group). (**K**) A representative photograph showing the traction of isolated ileum. (**L**) *Gcg* mRNA levels. (**M**) GLP-1 concentrations in the medium. (**N**) Western blot images and densitometry quantification for the indicated antibodies. Data are represented as mean ± SEM and are representative of six biological replicates. Significance was determined by Student's t test for comparison between two groups, and by one-way ANOVA for comparison among three groups or more, *p<0.05, **p<0.01, ***p<0.001.

The online version of this article includes the following source data for figure 4:

**Source data 1.** PDF file containing original western blots for *Figure 4F, J and N*, indicating the relevant bands and treatments.

**Source data 2.** Original files for western blot analysis displayed in *Figure 4F, J and N*.

**Source data 3.** Original data for *Figure 4*.

of GLP-1 (*Tolhurst et al., 2011*; *Xu et al., 2015*; *Yu and Jin, 2010*). Drawing from these findings, this research study proposed a hypothesis that Piezo1 may regulate GLP-1 synthesis via the CaMKKβ/CaMKIV-mTOR signaling pathway (*Figure 1N*). As shown in *Figure 1O*, abrogated GLP-1 production was associated with decreased CaMKKβ/CaMKIV-mTOR signaling in the ileal mucosa of *Piezo1* IntL-CKO mice (*Figure 1O*).

## Derangements of glucose metabolism and GLP-1 production were induced by HFD in *Piezo1* IntL-CKO mice, which was mitigated by Exendin-4

We next assessed the effect of L-cell-specific *Piezo1* gene deletion on GLP-1 and glucose tolerance in diet-induced diabetic mice. *Piezo1* IntL-CKO and control mice were exposed to HFD for 10 weeks. Compared to the controls, higher body weight (*Figure 2A*), greater glucose excursions (*Figure 2B*) were observed in *Piezo1* IntL-CKO mice exposed to HFD. Ileal mucosal Proglucagon expression levels were lower in *Piezo1* IntL-CKO than control mice (*Figure 2C–F*). Impaired CaMKKβ/CaMKIV-mTORC1 signaling pathway in ileal mucosa as evidenced by a decrease in CaMKKβ, reduced phosphorylation levels of CaMKIV, mTOR, S6K, and S6 was also observed in *Piezo1* IntL-CKO mice (*Figure 2F*). No significant alteration in morphology, Piezo1 or Proglucagon levels were observed in the pancreas of *Piezo1* IntL-CKO mice (*Figure 2—figure supplement 1A–D*). Together these data demonstrate that *Piezo1* IntL-CKO mice with prolonged HFD feeding exhibit impaired glucose metabolism phenotype and reduced GLP-1.

Injection of GLP-1 analog Exendin-4 (Ex-4) decreased the body weight (*Figure 2G*) and improved both glucose tolerance (*Figure 2H*) and insulin resistance (*Figure 2I*) in control and *Piezo1* IntL-CKO mice, while endogenous synthesis of GLP-1 was not changed by Ex-4 injection in *Piezo1* IntL-CKO mice (*Figure 2J and K*). These data suggested that decreased GLP-1 synthesis and secretion contribute to impaired glucose metabolism in *Piezo1* IntL-CKO mice.

## The pharmacological and mechanical activation of ileal Piezo1 stimulates GLP-1 synthesis

We next examined whether activation of Piezo1 could rescued the impaired glucose metabolism in diet-induced diabetic mice. Injection of Piezo1 activator Yoda1 after 10 weeks of high-fat diet, led to reduced body weight and improved the impaired glucose metabolism significantly in diabetic mice, while Piezo1 antagonist GsMTx4 reversed the weight loss and glucose-lowering effect of Yoda1 (*Figure 3A and B*). Yoda1 remarkably induced an increase in GLP-1 synthesis and secretion (*Figure 3C and D*), as well as an increment of CaMKKβ/CaMKIV-mTORC1 signaling in ileal mucosa (*Figure 3E*), while GsMTx4 abolished the effect of Yoda1 (*Figure 3C–E*). However, weight loss, improved plasma glucose and increased GLP-1 production induced by Yoda1 were not observed in *Piezo1* IntL-CKO mice (*Figure 3F–J*).

The intestine receives mechanical stimulation from the chyme, which may activate Piezo1 in the intestine epithelium, including L cells. To mimic the mechanical pressing and stretching induced by intestinal contents, a small silicon bead was implanted into the high-fat diet-induced diabetic mouse ileum. To exclude the possibility of bowel obstruction and abdominal pain caused by bead

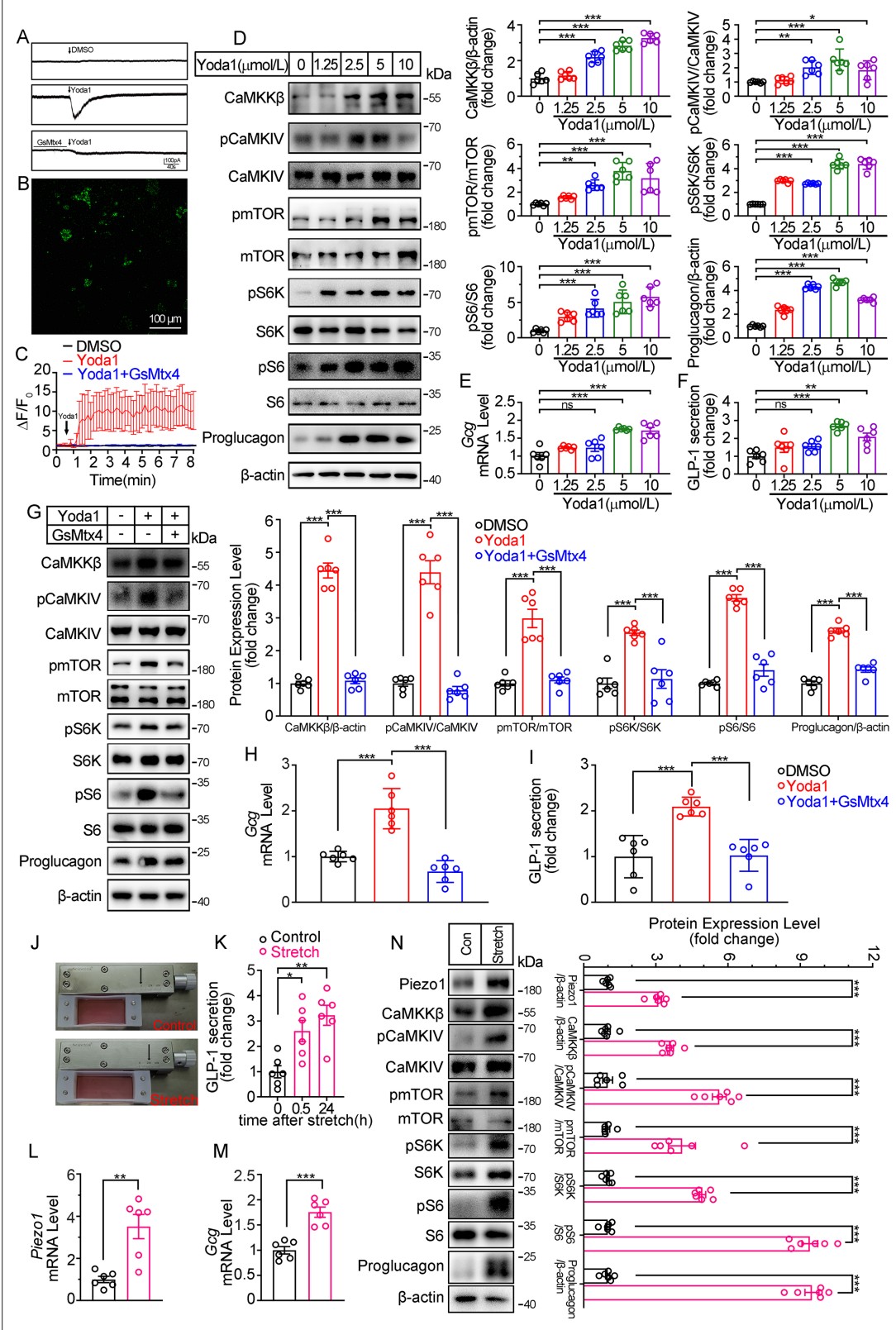

**Figure 5.** Modulation of GLP-1 synthesis and secretion by pharmacological and mechanical activation of Piezo1 in STC-1 cells. (**A**) Whole-cell currents induced by Yoda1 (5 μM) were recorded from STC-1 cells or STC-1 cells pretreated with GsMTx4 for 30 min. (**B, C**) Intracellular calcium imaging in STC-1 cells. (**B**) STC-1 cells were loaded with fluo-4 AM for 1 hr. The representative time-lapse image showing the intracellular Ca²⁺ signals. (**C**) The change of fluorescent intensity (ΔF/F0) was plotted against time. (**D–F**) STC-1 cells were treated with various concentrations of Yoda1 for 24 hr. (**D**) Whole-cell

*Figure 5 continued on next page*

*Figure 5 continued*

extracts underwent western blot with indicated antibodies. (**E**) *Gcg* mRNA levels. (**F**) GLP-1 concentrations in the culture medium. (**G–I**) STC-1 cells were treated with Yoda1 (5 µM) in the presence or absence of GsMTx4 (0.1 µM) for 24 hr. (**G**) Whole-cell extracts underwent western blot with indicated antibodies. (**H**) *Gcg* mRNA levels. (**I**) GLP-1 concentrations in the culture medium. (**J–N**) STC-1 were subjected to mechanical stretch. (**J**) STC-1 cells were cultured in elastic chambers and the chambers were subjected to mechanical stretch by 120% extension of their original length. (**K**) The medium GLP-1 concentrations were detected at indicated time. (**L**) *Piezo1* mRNA levels. (**M**) *Gcg* mRNA levels. (**N**) Whole-cell extracts underwent western blot with indicated antibodies. Data are represented as mean ± SEM and are representative of six biological replicates. Significance was determined by Student's t test for comparison between two groups, and by one-way ANOVA for comparison among three groups or more, *p<0.05, **p<0.01, ***p<0.001.

The online version of this article includes the following source data for figure 5:

**Source data 1.** PDF file containing original western blots for *Figure 5D, G and N*, indicating the relevant bands and treatments.

**Source data 2.** Original files for western blot analysis displayed in *Figure 5D, G and N*.

**Source data 3.** Original data for *Figure 5*.

implantation, we measured the fecal mass and gastrointestinal transit time, and accessed abdominal mechanical sensitivity in both sham and bead-implanted mice. As shown in *Figure 3—figure supplement 1A, B*, there was no significant difference in fecal mass and gastrointestinal transit time between the sham-operated mice and those implanted with beads. The results of abdominal mechanical sensitivity indicated that no difference in abdominal pain threshold was observed between sham and bead implanted mice (*Figure 3—figure supplement 1C*). Intestinal bead implantation improved the impaired glucose metabolism in diabetic mice (*Figure 3K and L*). Body weight loss, activated ileal mucosal CaMKKβ/CaMKIV-mTOR signaling, increased mRNA and protein levels of ileal mucosal Piezo1 and Proglucagon, as well as the circulating levels of GLP-1 were observed in diabetic mice after operation (*Figure 3M–R*). The above data suggest that mechanical stimuli induced by intestinal bead implantation activates ileal Piezo1 in diabetic mice, stimulating GLP-1 production via CaMKKβ/CaMKIV-mTOR signaling axis, thus improving glucose homeostasis.

## Piezo1 regulates GLP-1 synthesis and secretion in primary cultured mouse L cells and isolated mouse ileum

To obtain primary L cells, we isolated cell from the ileum of *Vil1^FLP^::Gcg^frtCre^-Rosa26^mT/mG^* mice, in which tdTomato expression switched to EGFP expression in L cells as shown in *Figure 1E*. EGFP-positive cells (mouse L cells) were then sorted from isolated single cells (*Figure 4A*). Immunofluorescence showed that the sorted EGFP⁺ cells were Piezo1 positive (*Figure 4B*).

Yoda1 at the dose of 5 µM triggered an increase in intracellular Ca²⁺ level in primary cultured mouse L cells, which was blocked by pre-incubation of cells with GsMTx4 (0.1 µM) for 15 min (*Figure 4C*). Yoda1 also stimulated Proglucagon expression and GLP-1 secretion, as well as CaMKKβ/CaMKIV-mTOR signaling pathway in primary cultured mouse L cells (*Figure 4D–F*). In contrast, knockdown of *Piezo1* by shRNA led to significant decrease in Proglucagon expression and GLP-1 secretion, as well as inhibition of CaMKKβ/CaMKIV/mTOR signaling pathway (*Figure 4G–J*).

Given the ability of Piezo1 in sensing mechanical force, tension of 1.5 g was applied to the isolated mouse ileum bathed in Tyrode's solution for four hours. Tension stimulated Proglucagon expression, GLP-1 secretion and activated CaMKKβ/CaMKIV-mTOR signaling pathway in the ileum of control mice, but not in *Piezo1* IntL-CKO mice (*Figure 4K–N*), suggesting the involvement of Piezo1 of the L cells in mediating the force-induced GLP-1 production and CaMKKβ/CaMKIV-mTOR signaling.

## Pharmacological, mechanical and genetic activation of Piezo1 stimulates GLP-1 synthesis and secretion in STC-1 cells

To further validate the role of Piezo1 in regulating GLP-1, we examined the effect of manipulating Piezo1 on GLP-1 production in an intestinal neuroendocrine cell line STC-1. Pharmacological activation of Piezo1 by Yoda1 triggered an inward current in STC-1 cell recorded by whole cell patch-clamp, which could be inhibited by pre-incubation of GsMTx4 (*Figure 5A*). Yoda1 also triggered an increase in intracellular Ca²⁺ level in STC-1 cells. Pre-incubation of cells with GsMTx4 (0.1 µM) for 15 min inhibited [Ca²⁺]ᵢ increase (*Figure 5B and C*). Yoda1 induced a concentration-dependent activation of CaMKKβ/CaMKIV-mTOR pathway and GLP-1 synthesis and secretion (*Figure 5D–F*). GsMTx4 blocked the effect of Yoda1 on STC-1 cells in both GLP-1 and CaMKKβ/CaMKIV-mTOR activation (*Figure 5G–I*).

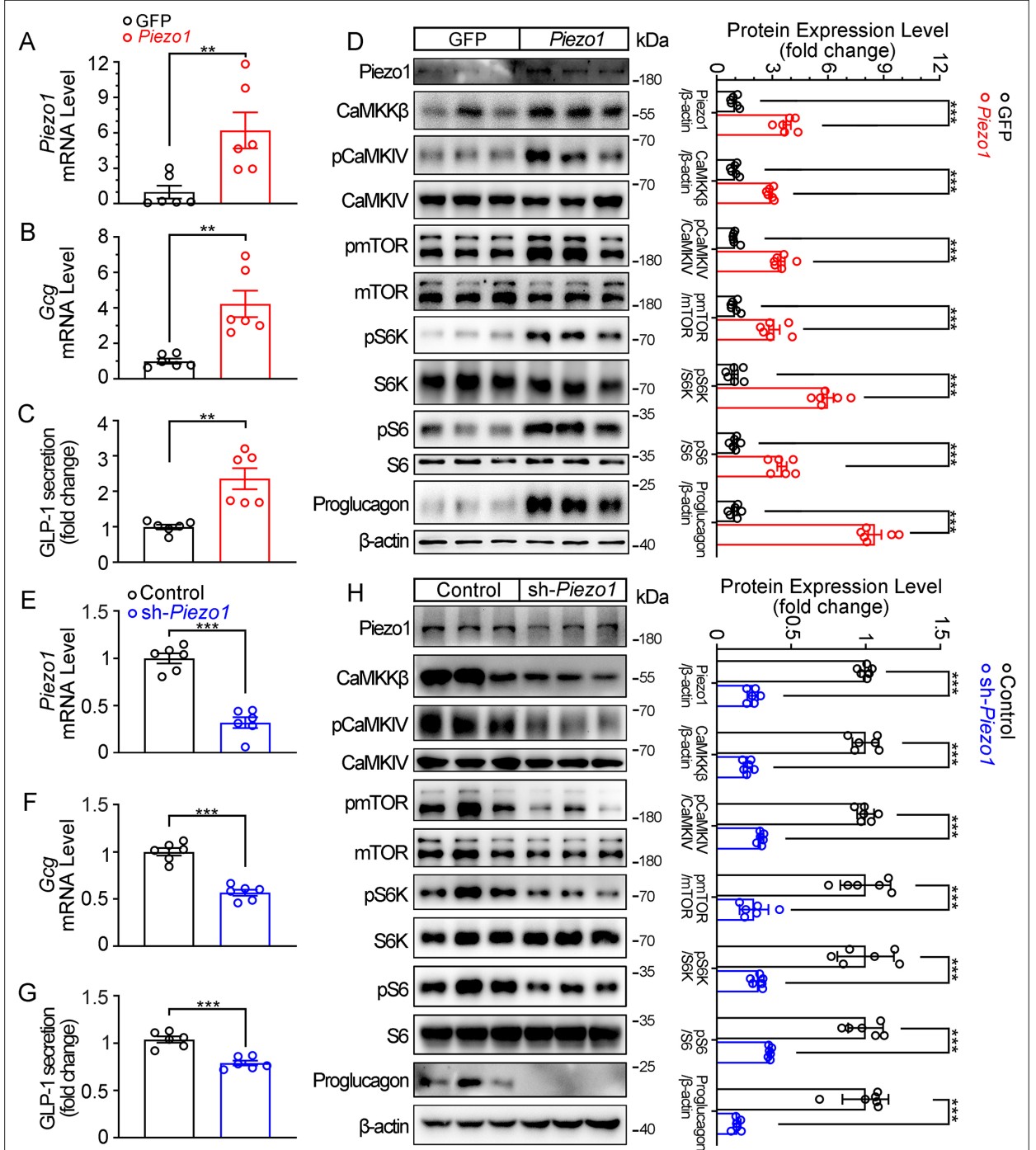

**Figure 6.** Genetic interference of Piezo1 regulates GLP-1 production in STC-1 cells. (**A–D**) STC-1 cells were transfected with mouse control or *Piezo1* expression plasmids for 48 hr. *Piezo1* (**A**) and *Gcg* (**B**) mRNA levels in STC-1 cells. (**C**) GLP-1 concentrations in culture medium. (**D**) Whole-cell extracts underwent western blot with indicated antibodies. (**E–H**) Stable knockdown of *Piezo1* in STC-1 cells. *Piezo1* (**E**) and *Gcg* (**F**) mRNA levels in STC-1 cells. (**G**) GLP-1 concentrations in culture medium. (**H**) Whole-cell extracts underwent western blot with indicated antibodies. Data are represented as mean ± SEM Data are represented as mean ± SEM and are representative of six biological replicates. Significance was determined by Student's t test, *p<0.05, **p<0.01, ***p<0.001.

The online version of this article includes the following source data for figure 6:

**Source data 1.** PDF file containing original western blots for ***Figure 6D and H***, indicating the relevant bands and treatments.

**Source data 2.** Original files for western blot analysis displayed in ***Figure 6D and H***.

**Source data 3.** Original data for ***Figure 6***.

To mimic the activation of Piezo1 by mechanical stretching in vivo, STC-1 cells grown on elastic chambers were subjected to mechanical stretch to 120% of their original length. Mechanical stretch upregulated Piezo1 and Proglucagon expression, promoted GLP-1 secretion (*Figure 5J–N*), and activated CaMKKβ/CaMKIV- mTOR signaling pathways (*Figure 5N*).

Consistent to the pharmacological and mechanical activation of Piezo1, over-expression of Piezo1 in STC-1 cells resulted in a significant increase in GLP-1 production, as well as activation of the CaMKKβ/CaMKIV-mTOR signaling pathway (*Figure 6A–D*). Conversely, knockdown of *Piezo1* by shRNA led to a significant decrease in GLP-1 production and inhibition of CaMKKβ/CaMKIV-mTOR signaling pathway (*Figure 6E–H*).

## Piezo1 regulates GLP-1 production through CaMKKβ/CaMKIV and mTOR in STC-1 cells

Next, we examined whether CaMKKβ/CaMKIV and mTOR signaling mediates the effects of Piezo1 on GLP-1 production. Overexpression of CaMKKβ or CaMKIV increased CaMKKβ/CaMKIV and mTOR signaling activity, resulting in increased synthesis and secretion of GLP-1 (*Figure 7A–C*). In contrast, the CaMKKβ inhibitor STO-609, downregulated CaMKKβ/CaMKIV and mTOR signaling, as well as GLP-1 synthesis and secretion (*Figure 7D–F*). Inhibition of mTORC1 activity by rapamycin suppressed GLP-1 production induced by Yoda1, which was associated with inhibition of mTOR signaling (*Figure 7G–I*).

## Discussion

It has been known for decades that GLP-1 secretion from the intestinal L cells is stimulated by meal intake and is essential for postprandial glycemic control (*Drucker, 2006*; *Song et al., 2019*). However, the mechanism underlying the regulation of GLP-1 production is not completely understood. One of the problems that impeded the investigation of regulation mechanism of GLP-1 is the lack of an L-cell-specific genetically engineered animal model. Here, an L-cell-specific Cre mouse line was generated for the first time through the combination of the FLP-Frt and Cre-LoxP systems. This enables genetic manipulation specifically in the L cells and creates a valuable tool for investigating molecular mechanisms in L cells.

Previous studies have shown that L cells are able to sense nutrients in the intestinal lumen such as glucose and other carbohydrates, lipids and amino acids, which induce GLP-1 secretion through different mechanisms, including membrane depolarization-associated exocytosis, $Ca^{2+}$/Calmodulin (*Tolhurst et al., 2011*), cAMP (*Yu and Jin, 2010*), mTORC1 (*Xu et al., 2015*), and AMPK (*Jiang et al., 2016*) signaling pathways. However, it is innegligible that as open type endocrine cells, L cells not only receive the chemical stimulations from the nutrients, but also mechanical stimulation when the chyme passing through the intestine, including stretching, pressure and shear force (*Sensoy, 2021*). While the food needs to be digested and nutrients absorbed before L-cells can detect the nutritive signals, mechanical stimulation may be more direct and faster. The expression of Piezo1, a mechanosensitive ion channel, was demonstrated in human and mouse intestinal sections, primary mouse L cell culture, and intestinal neuroendocrine cell line STC-1, indicating the mechanosensing ability of L cells and the potential regulatory effect of GLP-1 on mechanical stimulation. The results showed a significant increase in GLP-1 secretion by implantation of intestinal beads, stretching of intestinal tissue, or stretching of STC-1 cells, providing further evidence that mechanical regulation of GLP-1 secretion does exist. In addition, the selective deletion of Piezo1 (*Piezo1* IntL-CKO) mice showed reduced circulating GLP-1 level, increased body weight, and impaired glucose homeostasis, while pharmacological activation of Piezo1 in mice, primary L cells, and STC-1 cells showed opposite effects. More importantly, *Piezo1* IntL-CKO mice was unable to response to the tension-induced GLP-1 production. These further suggested a Piezo1-mediated mechanical sensing mechanism in L cells that regulates GLP-1 production and glucose metabolism by sensing the stimulation of intestinal luminal contents.

Interestingly, this intestinal Piezo1-mediated mechanical sensing mechanism may severely impaired in diabetic patients and rodents. Reduced expression of Piezo1 was demonstrated in the ileal mucosa of diet-induced diabetic mice, along with reduced GLP-1 production. When challenged with high-fat diet, *Piezo1* IntL-CKO mice exhibited more severe symptoms of diabetes which was mitigated by Ex-4. These findings suggest that the impairment of Piezo1-mediated mechanical sensing function in the intestine is an important mechanism for the pathogenesis of T2DM. It is noteworthy that RYGB,

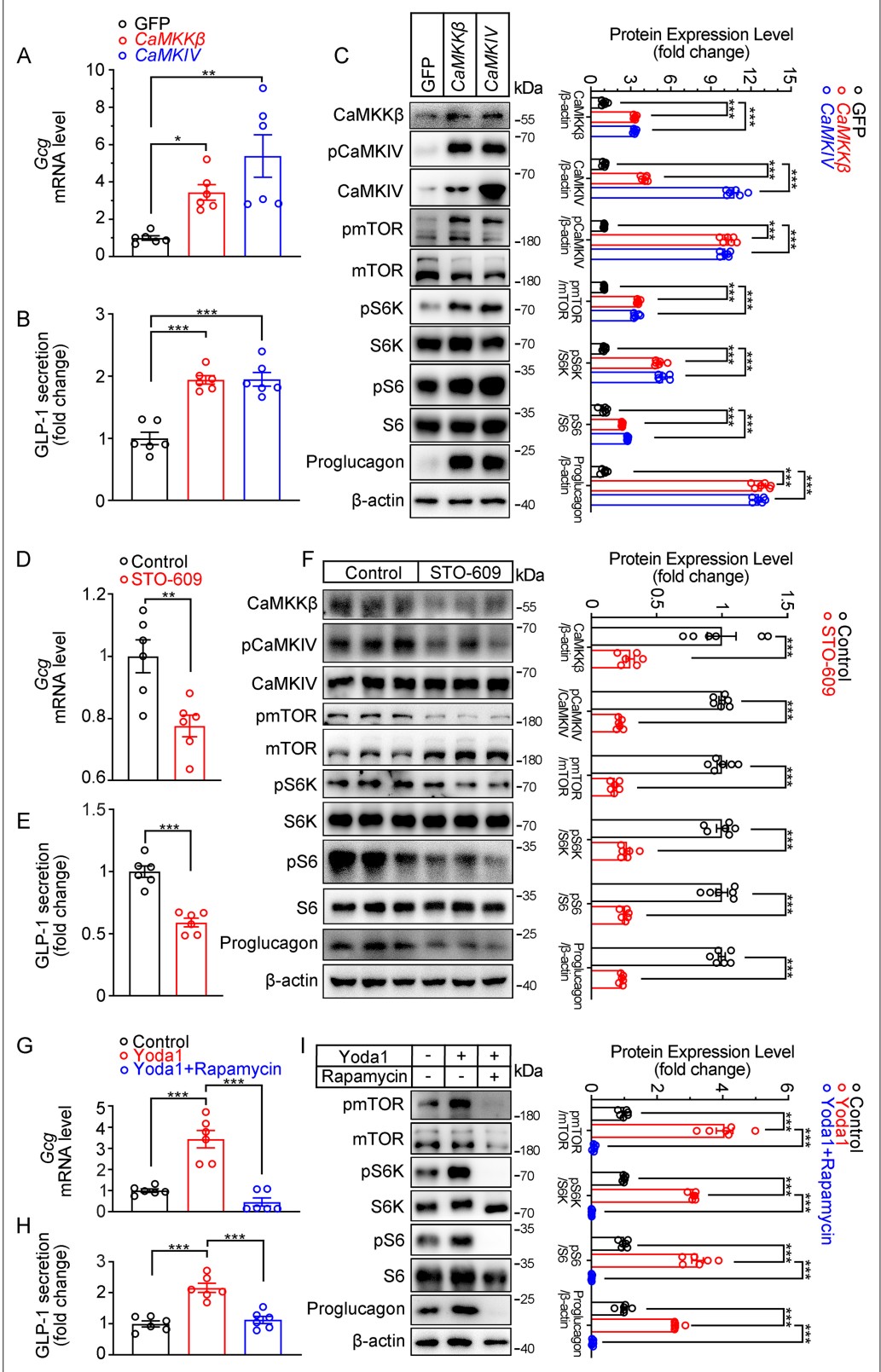

**Figure 7.** Modulation of GLP-1 production by CaMKKβ/CaMKIV and mTOR signaling activity in STC-1 cells.
(**A–C**) STC-1 cells were transfected with GFP, *CaMKKβ* or *CaMKIV* plasmids for 48 hr. (**A**) *Gcg* mRNA levels in STC-1 cells. (**B**) GLP-1 concentrations in culture medium. (**C**) Whole-cell extracts underwent western blot with indicated antibodies. (**D–F**) STC-1 cells were treated with CaMKKβ inhibitor STO-609 (10 µmol/L) for 24 hr. (**D**) *Gcg* mRNA

*Figure 7 continued on next page*

*Figure 7 continued*

levels in STC-1 cells. (**E**) GLP-1 concentrations in culture medium. (**F**) Whole-cell extracts underwent western blot with indicated antibodies. (**G–I**) STC-1 cells were pretreated with Rapamycin (50 nmol/L) for 1 hr, then treated with Yoda1 (5 µmol/L) for 24 hr. (**G**) *Gcg* mRNA levels in STC-1 cells. (**H**) GLP-1 concentrations in the culture medium. (**I**) Whole-cell extracts underwent western blot with indicated antibodies. Data are represented as mean ± SEM and are representative of six biological replicates. Significance was determined by Student's t test for comparison between two groups, and by one-way ANOVA for comparison among three groups or more, *p<0.05, **p<0.01, ***p<0.001.

The online version of this article includes the following source data for figure 7:

**Source data 1.** PDF file containing original western blots for **Figure 7C, F and I**, indicating the relevant bands and treatments.

**Source data 2.** Original files for western blot analysis displayed in **Figure 7C, F and I**.

**Source data 3.** Original data for **Figure 7**.

---

a commonly performed weight-loss and hypoglycemic surgery (**Cummings et al., 2004**), significantly increased Piezo1 expression in L cells of obese diabetic patients. Yoda1 treatment or intestinal bead implantation enhanced GLP-1 production and improved glucose metabolism in the diet-induced diabetic mouse model, suggesting that restoring the mechano-sensing or enhancing the function of Piezo1 either pharmacologically or mechanically, may be a new strategy to improve the secretion of GLP-1 and alleviate T2DM. However, various data suggest that Piezo1-mediated regulation of GLP-1 production has only been demonstrated in transgenic mice, mouse primary L cells, and enteric neuro-endocrine cell lines derived from mice. Whether Piezo1 plays the same role in human L cells awaits to be investigated. A number of studies have generated L cells culture from human intestinal organoid culture or human intestinal stem cell monolayer culture by manipulating the growth factors in the media (**Goldspink et al., 2020**; **Petersen et al., 2014**; **Villegas-Novoa et al., 2022**). It is worthy to validate our finding in human L cells in order to prove its translational potential in T2DM treatment.

The intragastric balloon is a current clinical weight loss measure that involves placing a space-occupying balloon in the stomach to reduce food intake and generate satiety signals, thus maintaining satiety. Investigations illustrated that intragastric balloon alter the secretion of hormones such as cholecystokinin and pancreatic polypeptide, delay the emptying of food in the stomach and reduce the appetite (**Mathus-Vliegen and de Groot, 2013**). Intragastric balloon provides a feasible weight loss intervention for obese people (**Kim et al., 2016**). In this study, a new intestinal implantation surgery of beads was adopted, which may offer a novel approach for weight loss and glucose control by activating the intestinal Piezo1-GLP-1 axis in the future.

Mechanistically, cellular and mouse models revealed that Piezo1 regulates GLP-1 production through the CaMKKβ/CaMKIV-mTOR signaling pathway. CaMKKβ/CaMKIV has been reported to mediate the $Ca^{2+}$ signaling in many metabolic processes, including liver gluconeogenesis and de novo lipogenesis, adipogenesis, insulin sensitivity, and β cell proliferation (**Anderson et al., 2012**; **Lin et al., 2011**; **Liu et al., 2012**; **Liu et al., 2022b**). mTOR plays a central role in nutrient and energy sensing and regulates cellular metabolism and growth in response to different nutrient and energy status (**Howell and Manning, 2011**). Here, the data suggest that mTOR can also response to mechanical stimuli through a mechano-sensitive $Ca^{2+}$ channel-mediated CaMKKβ/CaMKIV activation. Although it has not been demonstrated that CaMKIV directly phosphorylates mTOR or S6K in L cells, a previous study reported that CaMKKβ could serve as a scaffold to assemble CaMKIV with key components of the mTOR/S6K pathway and promote liver cancer cell growth (**Lin et al., 2015**), which lended support to the CaMKKβ/CaMKIV-mTOR signaling in our study. Recently, Knutson et. al. found that ryanodine and IP3-triggered calcium release from intracellular calcium store could amplified the initial Peizo2 - $Ca^{2+}$ signal triggered by mechanical stimulation, and was required for the mechanotrans-duction in the serotonin release from enterochromaffin cells (**Knutson et al., 2023**). Primary L-cell and STC-1 cell results showed a persistent intracellular $Ca^{2+}$ increase triggered by Yoda1, which also suggests that intracellular $Ca^{2+}$ stores are involved in $Ca^{2+}$ relay. Beside $Ca^{2+}$, cyclic AMP (cAMP) is another signaling molecule that active *Gcg* gene expression and GLP-1 production (**Drucker et al., 1994**; **Jin, 2008**; **Simpson et al., 2007**). cAMP was found to play a critical role in nutrients-induced GLP-1 secretion, including glucose (**Ong et al., 2009**), lipids (**Hodge et al., 2016**), and amino acids (**Tolhurst et al., 2011**). Previous study reported that $Ca^{2+}$ can activate soluble adenylyl cyclase (sAC)

to increase intracellular cAMP (*Jaiswal and Conti, 2003*). Whether sAC-cAMP can be activated by Piezo1-mediated $Ca^{2+}$ influx and whether it is an alternative signaling pathway that mediates the Piezo1-regulated GLP-1 production remain to be explored.

Furthermore, recent studies have highlighted the role of Piezo1 in enhancing insulin secretion (*Deivasikamani et al., 2019*; *Ye et al., 2022*), while inhibiting ghrelin (*Zhao et al., 2024*) and glucagon production (*Guo et al., 2024*), as well as reducing intestinal nutrient absorption (*Tao et al., 2024*). The diverse functions of Piezo1 across various cell types can be attributed to several factors, including cellular context, specific signaling pathways, and the microenvironment surrounding the cells. The current study reveals Piezo1-mediated mechanosensory properties of intestinal L cells that play an important role in regulating GLP-1 production and glucose metabolism. This finding suggests the existence of a new mechanoregulatory mechanism in enteroendocrine cells in addition to chemical and neural regulation, which may provide new ideas for the treatment of metabolic diseases such as diabetes and obesity.

# Materials and methods

**Key resources table**

| Reagent type (species) or resource | Designation | Source or reference | Identifiers | Additional information |
|---|---|---|---|---|
| Strain, strain background (*Mus musculus*, C57BL/6 J) | *Vil1^{FLP}*, *Gcg^{Cre}* | Shanghai Model Organisms Center | N/A | |
| Strain, strain background (*M. musculus*, C57BL/6 J) | *Vil1^{FLP}::Gcg^{frtCre}* | This paper | N/A | Please refer to the "Genetic mouse generation" section. |
| Strain, strain background (*M. musculus*, C57BL/6 J) | *Rosa26^{mTmG}* | Jackson Laboratory | Stock No. 007676 | |
| Strain, strain background (*M. musculus*, C57BL/6 J) | B6.Cg-Piezo1^{tm2.1Apat}/J | Jackson laboratory | RRID:IMSR_JAX:029213 | |
| Cell line (*M. musculus*, mouse) | STC-1 | ATCC | CRL-3254 | |
| Biological sample (*Mouse*) | Primary mouse ileal L cells, Ileum, Pancreas, Liver, Skeletal muscle, Epididymal adipose, Hypothalamus | This paper | N/A | Freshly isolated from Mice. |
| Transfected construct (*M. musculus*) | pLKO.1-shPiezo1 | This paper | N/A | Lentiviral construct to transfect and express the shRNA. |
| Antibody | Anti-Piezo1 (Rabbit polyclonal) | Affinity Biosciences | Cat# DF12083, RRID:AB_2844888 | WB: 1:1000 IF: 1:400 |
| Antibody | Anti-CaMKKβ (mouse monoclonal) | Santa Cruz Biotechnology | Cat# sc-271674, RRID:AB_10708844 | WB: 1:1000 |
| Antibody | Anti-Phospho-CaMKIV (Thr200) (Rabbit polyclonal) | Affinity Biosciences | Cat# AF3460, RRID:AB_2834898 | WB: 1:1000 |
| Antibody | Anti-CaMKIV (Rabbit polyclonal) | Cell Signaling Technology | Cat# 4032, RRID:AB_2068389 | WB: 1:1000 |
| Antibody | Anti-Phospho- mTOR (Ser2448) (Rabbit Monoclonal) | Cell Signaling Technology | Cat# 5536, RRID:AB_10691552 | WB: 1:1000 |
| Antibody | Anti-mTOR (Rabbit monoclonal) | Cell Signaling Technology | Cat# 2983, RRID:AB_2105622 | WB: 1:1000 |

*Continued on next page*

*Continued*

| Reagent type (species) or resource | Designation | Source or reference | Identifiers | Additional information |
|---|---|---|---|---|
| Antibody | Anti-phospho-p70 S6 Kinase (Thr389) (Rabbit monoclonal) | Cell Signaling Technology | Cat# 9234, RRID:AB_2269803 | WB: 1:1000 |
| Antibody | Anti-p70 S6 Kinase (Rabbit Monoclonal) | Cell Signaling Technology | Cat# 2903, RRID:AB_1196657 | WB: 1:1000 |
| Antibody | Anti-phospho-S6 Ribosomal Protein (Ser235/236) (Rabbit Monoclonal) | Cell Signaling Technology | Cat# 4858, RRID:AB_916156 | WB: 1:1000 |
| Antibody | Anti-S6 Ribosomal Protein (Rabbit monoclonal) | Cell Signaling Technology | Cat# 2217, RRID:AB_331355 | WB: 1:1000 |
| Antibody | Anti-GLP-1 (Mouse monoclonal) | Abcam | Cat# ab23468, RRID:AB_470325 | WB: 1:1000 IF: 1:500 |
| Antibody | Anti-β-actin (Mouse monoclonal) | Cell Signaling Technology | Cat# 3700, RRID:AB_2242334 | WB: 1:1000 |
| Antibody | Horseradish peroxidase-conjugated, Goat Anti-Rabbit IgG | Jackson ImmunoResearch Labs | Cat# 111-035-003, RRID:AB_2313567 | 1:10,000 |
| Antibody | Horseradish peroxidase-conjugated, Goat Anti-Mouse IgG | Jackson ImmunoResearch Labs | Cat# 115-035-003, RRID:AB_10015289 | 1:10,000 |
| Antibody | Goat anti-mouse fluorescein isothiocyanate-conjugated IgG | EarthOx LLC | Cat# E031210-01 | 1:100 |
| Antibody | Dylight 594 affinipure donkey anti-rabbit IgG | EarthOx LLC | Cat# E032421-01 | 1:100 |
| Recombinant DNA reagent | pcDNA3.1-mPiezo1-IRES-GFP | Addgene | Cat# 80925 | |
| Recombinant DNA reagent | pcDNA3.1-IRES-GFP | Addgene | Cat# 51406 | |
| Recombinant DNA reagent | CaMKKβ (Plasmid) | This paper | N/A | Gifted by Professor Koji Murao from Kagawa University |
| Recombinant DNA reagent | CaMKIV (Plasmid) | This paper | N/A | Gifted by Professor Koji Murao from Kagawa University |
| Sequence-based reagent | P1 | This paper | PCR primers | GACCTTTGCCCTCTGGTCTC |
| Sequence-based reagent | P2 | This paper | PCR primers | GAGTGACGGTGCCAGAGAAA |
| Sequence-based reagent | P3 | This paper | PCR primers | GACTCCAGCTGCCTTCTCTG |
| Sequence-based reagent | P4 | This paper | PCR primers | CGGTGATCTCCCAGATGCTC |
| Sequence-based reagent | P5 | This paper | PCR primers | CCCTAACTCAGTCTCCAGCA |
| Sequence-based reagent | P6 | This paper | PCR primers | CGGTTACCAGGTGGTCATGT |
| Sequence-based reagent | P7 | This paper | PCR primers | CCCTAACTCAGTCTCCAGCA |

*Continued on next page*

*Continued*

| Reagent type (species) or resource | Designation | Source or reference | Identifiers | Additional information |
|---|---|---|---|---|
| Sequence-based reagent | P8 | This paper | PCR primers | CTGCAAAGGGTCGCTACAGA |
| Sequence-based reagent | P9 | This paper | PCR primers | AATGGCTCTCCTCAAGCGTAT |
| Sequence-based reagent | P10 | This paper | PCR primers | ACAGGAGGTAGTCCCTCACAT |
| Sequence-based reagent | P11 | This paper | PCR primers | TGTCGGGGAAATCATCGTCC |
| Sequence-based reagent | *Piezo1*_F (Human) | This paper | PCR primers | ATCGCCATCATCTGGTTCCC |
| Sequence-based reagent | *Piezo1*_R (Human) | This paper | PCR primers | TGGTGAACAGCGGCTCATAG |
| Sequence-based reagent | *GCG*_F (Human) | This paper | PCR primers | GCACATTCACCAGTGACTACAGCA |
| Sequence-based reagent | *GCG*_R (Human) | This paper | PCR primers | TGGCAGCTTGGCCTTCCAAATA |
| Sequence-based reagent | β-actin_F (Human) | This paper | PCR primers | TCATGAAGATCCTCACCGAG |
| Sequence-based reagent | β-actin_R (Human) | This paper | PCR primers | CATCTCTTGCTCGAAGTCCA |
| Sequence-based reagent | *Piezo1*_F (Mouse) | This paper | PCR primers | GCAGTGGCAGTGAGGAGATT |
| Sequence-based reagent | *Piezo1*_R (Mouse) | This paper | PCR primers | GATATGCAGGCGCCTATCCA |
| Sequence-based reagent | *Gcg*_F (Mouse) | This paper | PCR primers | ATTGCCAAACGTCATGATGA |
| Sequence-based reagent | *Gcg*_R (Mouse) | This paper | PCR primers | GGCGACTTCTTCTGGGAAGT |
| Sequence-based reagent | *CCK*_F (Mouse) | This paper | PCR primers | TAGCGCGATACATCCAGCAGGT |
| Sequence-based reagent | *CCK*_R (Mouse) | This paper | PCR primers | GGTATTCGTAGTCCTCGGCACT |
| Sequence-based reagent | *Actb*_F (Mouse) | This paper | PCR primers | CCACAGCTGAGAGGGAAATC |
| Sequence-based reagent | *Actb*_R (Mouse) | This paper | PCR primers | AAGGAAGGCTGGAAAAGAGC |
| Commercial assay or kit | Mouse Glucagon-Like Peptide 1 (GLP-1) ELISA Kit | Millipore | Cat# EGLP-35K | Mouse Glucagon-Like Peptide 1 (GLP-1) ELISA Kit |
| Commercial assay or kit | RT-PCR kit | Takara | Cat# RR014A | RT-PCR kit |
| Chemical compound, drug | 0.1% gelatine | Biological Industries | Cat# 01-944-1B | |
| Chemical compound, drug | DMEM high sugar medium | Gibco | Cat# 11965092 | |
| Chemical compound, drug | Fetal bovine serum | Gibco | Cat# 12484028 | |

*Continued*

| Reagent type (species) or resource | Designation | Source or reference | Identifiers | Additional information |
|---|---|---|---|---|
| Chemical compound, drug | Equine serum | Gibco | Cat# 16050122 | |
| Chemical compound, drug | Immobilon western chemiluminescent HRP substrate | Millipore | Cat# WBKLS0500 | |
| Chemical compound, drug | Diprotin A | Sigma-Aldrich | Cat# 90614-48-5 | |
| Chemical compound, drug | Thermo Scientific TurboFect Transfection Reagent | Thermo Fisher Scientific | Cat# R0531 | |
| Chemical compound, drug | TRIzol | Thermo Fisher Scientific | Cat# 15596026 | |
| Chemical compound, drug | RIPA Lysis Buffer | Beyotime Biotechnology | Cat# P0013B | |
| Chemical compound, drug | GsMTx4 | Alomone Labs | Cat# STG-100 | |
| Chemical compound, drug | Rapamycin | Santa Cruz Biotechnology | Cat# sc-3504B | |
| Chemical compound, drug | STO-609 | Selleck | Cat# S8274 | |
| Chemical compound, drug | Yoda1 | Sigma-Aldrich | Cat# SML1558 | |
| Chemical compound, drug | Dimethyl sulfoxide | Sigma-Aldrich | Cat# D2650 | |
| Chemical compound, drug | Exendin-4 | Sigma-Aldrich | Cat# E7144 | |
| Chemical compound, drug | Fluo-4 AM | Thermo Fisher Scientific | Cat# F14201 | |
| Software, algorithm | GraphPad Prism | GraphPad Software, https://www.graphpad.com/ | RRID:SCR_002798 | |
| Software, algorithm | ImageJ | ImageJ, https://imagej.nih.gov/ij/ | RRID:SCR_003070 | |
| Software, algorithm | Adobe photoshop | Adobe, https://www.adobe.com/creativecloud/desktop-app.html | RRID:SCR_014199 | |
| Other | Normal chow diet | Research Diets | Cat# D12450B | Feed for feeding mice. |
| Other | High fat diet | Research Diets | Cat# D12492 | Feed for feeding mice. |

## Collection of human intestine samples

Male obese participants with type 2 diabetes (n=6, BMI = 45.87 ± 4.889 kg/m$^2$) and one-year post-RYGB patients (n=6, BMI = 25.48 ± 1.085 kg/m$^2$) were recruited in current study. Written informed consent was obtained from each donor. The study protocol was approved by the Institutional Review Board of Jinan University. Mucosal biopsies were obtained from human intestines by using a colonoscopy (CF-HQ290I; Olympus).

## Genetic mouse generation

### Vil1<sup>FLP</sup> mice

*Vil1<sup>FLP</sup>* knock-in mouse model was developed by Shanghai Model Organisms Center, Inc. The targeting construct was designed to insert a 2A-Flp-WPRE-pA coexpression cassette into the stop codon of mouse *Vil1* gene via homologous recombination using CRISPR/Cas9 system. 5'-AGCCCCTACCCT GCCTTCAA-3' was chosen as Cas9 targeted guide RNA (sgRNA). The donor vector, sgRNA and Cas9 mRNA was microinjected into C57BL/6 J fertilized eggs. F0 generation mice positive for homologous recombination were identified by long PCR. The primers (I-IV) used for detection of the correct homology recombination were I: 5'-ACTTCAGGCCTAACGCTCAC-3' and II: 5'-TGTCCTGCAGGC AGAGAAAG-3' for the correct 5' homology arm recombination, and III: 5'-GTGCCGTCTCTAAGCA CAGT-3'and IV: 5'-TGTTGGTGCTTCGGAGTGTT-3'for the correct 3' homology arm recombination. The PCR products were further confirmed by sequencing. F0 mice were crossed with C57BL/6 J mice to obtain *Vil1<sup>FLP</sup>* heterozygous mice.

### FLP-dependent glucagon-Cre (Gcg<sup>Cre</sup>) mice

*Gcg<sup>Cre</sup>* mouse model was developed by Shanghai Model Organisms Center, Inc. The targeting construct was designed to insert an IRES-F3-Frt-Wpre-pA-Cre-Frt-F3 expression cassette into the 3' UTR of mouse *Gcg* gene of via homologous recombination using CRISPR/Cas9 system. 5'-ATGCAAAG CAATATAGCTTC-3' was chosen as Cas9 targeted guide RNA (sgRNA). The donor vector, sgRNA and Cas9 mRNA was microinjected into C57BL/6 J fertilized eggs. F0 generation mice positive for homologous recombination were identified by long PCR. The primers (I-IV) used for detection of the correct homology recombination were I: 5'-TGCTACACAGGAGGTCTGTC-3' and II: 5'-AGGCATGCTCTG CTATCACG-3' for the correct 5' homology arm recombination, and III: 5'-CCCTCCTAGTCCCTTC TCAGT-3' and IV: 5'-GCCAAGGACATCTTCAGCGA-3' for the correct 3' homology arm recombination. The PCR products were further confirmed by sequencing. F0 mice were crossed with C57BL/6 J mice to obtain *Gcg<sup>cre</sup>* heterozygous mice.

### Vil1<sup>FLP</sup>::Gcg<sup>frtCre</sup> mice

*Vil1<sup>FLP</sup>* mice were crossed with *Gcg<sup>cre</sup>* mice to obtain Intestinal L cell-specific Cre (*Vil1<sup>FLP</sup>::Gcg<sup>frtCre</sup>*) mice.

### *Piezo1* IntL-CKO *mice*

*Piezo1<sup>loxp/loxp</sup>* mice (B6.Cg-Piezo1<sup>tm2.1Apat</sup>/J) purchased from Jackson laboratory were crossed with *Vil1<sup>FLP</sup>::Gcg<sup>frtCre</sup>* mice to generate *Piezo1* IntL-CKO mice.

PCR is used to identify the genotype of mice during the subsequent mating and breeding process. The primers required for mouse genotyping are shown in the Key Resources Table.

## Mouse validation

*Rosa26<sup>mT/mG</sup>* reporter mice were purchased from Jackson laboratory, Inc *Vil1<sup>FLP</sup>::Gcg<sup>frtCre</sup>* mice were bred with *Rosa26<sup>mT/mG</sup>* reporter mice to further validate Cre recombinase activity and specificity. Every single *Vil1<sup>FLP</sup>::Gcg<sup>frtCre</sup>* mouse was confirmed by *Rosa26<sup>mT/mG</sup>* reporter mice before breeding with *Piezo1<sup>loxp/loxp</sup>* mice to generate *Piezo1* IntL-CKO mice.

## Frozen tissue confocal imaging

*Vil1<sup>FLP</sup>::Gcg<sup>frtCre</sup>-Rosa26<sup>mT/mG</sup>* reporter mice were sacrificed. Fresh ileum and pancreas tissues were collected and embedded in O.C.T. compound for histological analysis immediately. Slices of the tissues were cut for confocal imaging, which was protected from light. Fluorescence was detected by laser scanning confocal microscopy (*Li et al., 2022*).

## Animal housing and treatment

Male mice were maintained on a 12 hr light/12 hr dark cycle environment. Normal chow diet (NCD) or a high-fat diet (HFD) and water were available ad libitum unless specified otherwise. The animal protocols were approved by the Animal Care and Use Committee of Jinan University (IACUC-20230517–01).

Male *Piezo1* IntL-CKO mice and age-matched control littermates (*Piezo1<sup>loxp/loxp</sup>*, *Vil1<sup>FLP</sup>*, *Gcg<sup>Cre</sup>*, *Vil1<sup>FLP</sup>::Gcg<sup>frtCre</sup>* mice) fed with NCD or HFD were used in the experiments.

Male *Piezo1*^loxp/loxp and *Piezo1* IntL-CKO mice fed with 10 week-high fat diet were intraperitoneally injected with normal saline (NS) or the GLP-1R agonist Exendin-4 (100 µg/kg body weight) for 7 consecutive days.

High fat diet induced diabetic mice were randomly divided into 3 groups. When indicated, animals were injected intraperitoneally with Vehicle, Yoda1 (2 µg per mouse) or GsMTx4 (250 µg/kg) plus Yoda1 for 7 consecutive days.

High fat diet treated *Piezo1* IntL-CKO mice were randomly divided into 2 groups. When indicated, animals were injected intraperitoneally with Vehicle, Yoda1 (2 µg per mouse) for 7 consecutive days.

Diet induce diabetic C57BL/6 J mice were divided into sham and intestinal bead implantation groups.

## Food and water intake detection

The food and water intake were quantified using metabolic cages (Cat 41853, Ugo Basile, Comerio, Italy). The mice were individually housed in these specialized cages and given a period of 3 days to acclimate before data collection began. They had unrestricted access to food and water, which was continuously monitored throughout the study. The 41850–010 software/interface package, consisting of EXPEDATA (for data analysis) and METASCREEN (for data collection) software, along with the IM-2 interface module, was employed to record and analyze the data.

## Intraperitoneal glucose tolerance test

Mice were fasted for 12 hr before measuring their fasting glucose levels. An intraperitoneal glucose tolerance test (IPGTT) was performed by administering 1.5 g/kg body weight of glucose. Blood glucose concentrations were measured at specified time points using a glucometer by collecting tail vein blood samples.

## Insulin tolerance test

Mice were subjected to a 4 hr fast before measurement of fasting glucose were taken. Insulin tolerance tests (ITT) were conducted with a dose of 0.75 U/kg body weight of insulin. Blood glucose levels were measured at specified time points.

## Intestinal bead implantation

High-fat diet-induced type 2 diabetic C57BL/6 J mice were fasted 6–8 hr before the operation. Standard aseptic procedures were used throughout the operation. Intestinal bead implantation was similar to gastric bead implantation described in our previous study (*Zhao et al., 2024*). Briefly, a 1 cm incision was made on the abdominal wall to expose the intestine. A 1 cm incision was made approximately 1 cm above the ileocecal region. A 2.5 mm diameter bead was implanted into the ileum of the mouse through an incision. Then the wound was closed with suture. Finally, the abdominal wall was closed with suture. For sham operation, all the procedures were the same as the bead implantation except that the bead was not implanted.

## Detection of abdominal mechanical sensitivity

The mice were familiarized with a metal mesh floor covered with plastic boxes for 2 hr each day for 2 days prior to testing. Their abdominal fur was shaved 1 day before the experiments. The abdominal area was then stimulated using calibrated von Frey filaments (VFFs) that applied varying forces: subthreshold mechanical stimuli (indicative of allodynia, 0.07 g force) and painful stimuli (indicative of hyperalgesia, 0.16 and 1 g force). Each filament was applied 10 times for 5–8 s with 10 s intervals between applications. To prevent learning or sensitization, the same area was not stimulated more than once consecutively. The data were presented as the number of withdrawal responses out of 10 applications, with 0 indicating no withdrawal and 10 indicating the maximum withdrawal. A withdrawal response was defined as (1) the mouse withdrawing its abdomen from the VFFs, (2) subsequent licking of the abdominal area, or (3) withdrawal of the entire body. All tests were conducted in a blinded manner.

## Gastrointestinal transit time

The whole-gut transit time test was conducted as previously described (*Qin et al., 2017*). The duration between the oral administration of charcoal and the appearance of the first stained fecal pellet was recorded as the total gastrointestinal transit time.

## Stretching of isolated ileum

About 2 cm ileum was isolated from control and *Piezo1* IntL-CKO mice and kept in the specimen chamber filled with Tyrode's solution (KCl 0.2 g/L, NaCl 8 g/L, CaCl$_2$ 0.2 g/L, MgCl$_2$ 0.1 g/L, NaHCO$_3$ 1 g/L, NaH$_2$PO$_4$ 0.05 g/L, Glucose 1 g/L) of 37°C gassed with oxygen. The specimen was connected to the force transducer of organ bath system (HW200S, Techman, Chengdu, CN). Adjust the transducer to apply traction force of 1.5 g on the tissue and maintained for 4 hr.

## Measurement of GLP-1 secretion

The measurement of GLP-1 secretion was carried out according to previously described methods (*Zhai et al., 2018*). Samples were collected in the presence of aprotinin (2 µg/mL), EDTA (1 mg/mL) and diprotin A (0.1 mmol/L), and stored at –80 °C before use. GLP-1 levels were assayed using enzyme immunoassay kits following the manufacturer's instructions.

## Histological analysis

Tissues were collected, fixed with 4% paraformaldehyde, embedded in paraffin, and cut into 4 µm sections. Standard protocols were followed for staining the sections with hematoxylin-eosin. Photomicrographs were captured under an inverted microscope (Leica, Germany).

## Immunofluorescence

Paraffin-embedded tissue sections were dewaxed and rehydrated, followed by boiling in 0.01 mol/L citrate buffer (pH 6.0) for 10 min. Subsequently, the sections were blocked with 5% goat serum for 1 hr and then incubated overnight at 4 °C with the following primary antibodies: Piezo1 (1:400), Glucagon (1:200), Ghrelin (1:100), GLP-1(1:500), PYY (1:100), ZO-1 (1:200), or Occludin (1:200). The sections were then incubated with a mixture of secondary antibodies. Images were taken by laser scanning confocal microscopy (Leica SP8). Fluorescence intensity was quantified by ImageJ software.

## In situ hybridization

Paraffin sections were dewaxed and rehydrated. After antigen retrieval in in citrate buffer (pH6.0), the sections were incubated with Proteinase K (5 µg/ml) at 37 °C for the 15 min. Then the sections were hybridized with the probes overnight in a temperature-controlled chamber at 40 °C. The Piezo1 probe sequences were as follows: 5'-CTGCAGGTGGTTCTGGATATAGCCC-3',5'-AAGAAGCAGATC TCCAGCCCGAAT-3', 5'-GCCATGGATAGTCAATGCACAGTGC-3'. After washing with SSC buffers, the sections were hybridized in pre-warmed branch probes at 40 °C for 45 min. After washing with SSC buffers, the sections were hybridized with signal probe at 42 °C for 3 hr. After washing with SSC buffers, the sections were blocked with normal serum and then incubated with mouse anti-GLP-1 (1:500) antibody at 4 °C overnight followed by secondary antibody. Images were taken laser scanning confocal microscopy and the fluorescence signals were quantified by ImageJ.

## Western blot analysis

Tissues and cells were harvested. Ileal mucosa was scraped for protein extraction. Protein extraction was performed by using RIPA lysis buffer (50 mM Tris PH 7.4, 150 mM NaCl, 1% Triton X-100, 0.1% SDS, 1% Sodium deoxycholate, 1 mM PMSF and protease inhibitor cocktail.), then 40 µg of proteins were loaded onto an SDS-PAGE gel for separation. After the separation, the proteins were transferred onto a nitrocellulose membrane. The membrane was then incubated in blocking buffer at room temperature for 1 hur. For overnight incubation, the membrane and primary antibody (at the recommended dilution as stated in the product datasheet) were immersed in primary antibody dilution buffer, with gentle agitation, at 4 °C. Subsequently, the membrane was incubated with a secondary antibody that specifically recognizes and binds to the primary antibody. Finally, western blotting luminol reagent was used to visualize bands. The grey scale values of the bands were measured using ImageJ software.

## RNA extraction, quantitative real-time PCR

RNA was extracted and reverse-transcribed into cDNAs using RT-PCR kit. Real-time PCR was performed as previously described (*Zhai et al., 2018*). Sequences for the primer pairs used in this study were shown in Key Resources Table.

## Isolation of mouse intestinal L cells

A 5~6 cm long ileum segment was collected from the *Vil1^FLP^::Gcg^frtCre^-Rosa26^mT/mG^* mouse. The tissue was washed with ice-cold PBS twice to remove the chyme in the lumen. The tissue was minced into $0.5 \text{ mm}^3$ pieces in ice-cold PBS and then digested in 100mIU collagenase I and 0.01 g/mL trypsin at 37 °C for 30 min with rotation. After digested tissue was passed through 40 µm and 30 µm cell strainers sequentially, then centrifuged for 7 min at 4 °C. The cell pellet was resuspended in red cell lysis buffer and incubated for 10 min at room temperature. The unlysed cells were collected by centrifugation and resuspended with 1 mL cold PBS. The GFP positive cells was sorted by fluorescence-activated cell sorting (FASC) on Beckman Coulter MoFlo XDP cell sorter system.

## Cell culture and treatments

STC-1 cells were maintained in DMEM medium supplemented with 2.5% fetal bovine serum and 10% equine serum at 37 °C with 5% $CO_2$ air. L cells were maintained in DMEM medium supplemented with 10% fetal bovine serum.

For cell transfection, cells were plated at optimal densities and grown for 48 hr. Cells were then transfected with *GFP*, *Piezo1* (Addgene, MA, USA), *CaMKKβ* and *CaMKIV* constructs (*Murao et al., 2009*) by using lipofectamine reagent according to the manufacturer's instructions.

For stable knockdown of Piezo1 in STC-1 cells, short hairpin RNA (shRNA) sequences for mouse Piezo1 interference were cloned in to pLKO.1 vector. To produce lentivirus, psPAX2, pMD2G and pLKO.1 or pLKO.1-shPiezo1 plasmids (siPiezo1: CCAACCTTATCAGTGACTT) were co-transfected into 293T cells with lipofectamine 2000 reagent. Supernatant containing lentivirus was collected 48 hr after transfection and filtered through 0.45 µm filter. The virus-containing supernatant was used to infect STC-1 cells. Forty-eight hours after infection, the STC-1 cells were subjected to 1 µg/mL puromycin selection for 2–3 days.

For cell stretching, cells were grown in silicone elastic chambers coated by 0.1% gelatin solution. After incubated at 37 °C for 24–48 hr, The chambers were subjected to mechanical stretch to 120% of their original length.

## Calcium imaging

Cells were plated onto confocal dishes at optimal densities and grown for 24 hr. Cells were loaded with the calcium fluorescent probe fluo-4 AM (1 µM) for 1 hr at 37 °C, then the cells were treated with Yoda1 (5 µM) or GsMTx4. The intracellular calcium ions were measured at room temperature using a laser confocal microscope with an excitation wave length of 494 nm and an emission wave length of 516 nm. The change of fluorescent signal was presented as $\Delta F/F_0$ and plotted against time.

## Whole-cell patch-clamp recording

Borosilicate glass-made patch pipettes (BF150-86-7.5, Sutter Instrument Co, USA) were pulled with micropipette puller (P-1000, Sutter Instrument Co, USA) to a resistance of 3–5 MΩ after being filled with pipette solution: 138 mM KCl, 10 mM NaCl, 1 mM $MgCl_2$, 10 mM Glucose and 10 mM HEPES (pH 7.4). Cells were bathed in Margo-Ringer solution: 130 mM NaCl, 5 mM KCl, 1 mM $MgCl_2$, 2.5 mM $CaCl_2$, 10 mM Glucose, 20 mM HEPES (pH7.4). Whole-cell calcium currents of STC-1 cells were recorded with the EPC10 USB patch-clamp amplifier (HEKA, Germany) controlled by PatchMaster software.

## Statistical analysis

All data were expressed as mean ± S.E.M. Statistical differences were evaluated by one-way ANOVA or Student's t-test. The correlation was determined by Pearson analysis. $p < 0.05$ was considered significant. (*$p < 0.05$, **$p < 0.01$, *** $p < 0.001$, ns = not significance). In this study, the data collection and analysis processes were not conducted in a blinded manner with respect to the experimental conditions. For the administration of drugs to animals, we allocated mice of the same genetic background

to various experimental cohorts using a randomization protocol. No data were excluded during the data analysis.

## Acknowledgements

This work was supported by grants from the National Natural Science Foundation of China (82170818, 81770794), Guangdong Basic and Applied Basic Research Foundation (2024A1515010686), the Fundamental Research Funds for the Central Universities (21620423), Guangzhou Science and Technology Plan Project Funding (202201011353).

## Additional information

### Funding

| Funder | Grant reference number | Author |
| --- | --- | --- |
| National Natural Science Foundation of China | 82170818 | Geyang Xu |
| National Natural Science Foundation of China | 81770794 | Geyang Xu |
| Guangdong Basic and Applied Basic Research Foundation | 2024A1515010686 | Geyang Xu |
| Fundamental Research Funds for the Central Universities | 21620423 | Geyang Xu |
| Guangzhou Science and Technology Plan Project Funding | 202201011353 | Jie Yang |

The funders had no role in study design, data collection and interpretation, or the decision to submit the work for publication.

### Author contributions

Yanling Huang, Haocong Mo, Data curation, Software, Formal analysis, Validation, Investigation, Methodology, Writing - original draft; Jie Yang, Data curation, Formal analysis, Funding acquisition, Validation, Investigation, Methodology; Luyang Gao, Data curation, Formal analysis, Validation, Investigation, Methodology, Writing - original draft; Tian Tao, Data curation, Investigation, Methodology; Qing Shu, Wenying Guo, Yawen Zhao, Formal analysis, Validation, Investigation; Jingya Lyu, Resources, Formal analysis, Validation; Qimeng Wang, Formal analysis, Validation; Jinghui Guo, Hening Zhai, Resources, Investigation; Linyan Zhu, Resources, Investigation, Methodology; Hui Chen, Conceptualization, Resources, Software, Formal analysis, Supervision, Investigation, Visualization, Methodology, Writing – review and editing; Geyang Xu, Conceptualization, Resources, Formal analysis, Supervision, Funding acquisition, Investigation, Visualization, Methodology, Project administration, Writing – review and editing

### Author ORCIDs

Haocong Mo ⓘ https://orcid.org/0009-0009-4778-8062
Luyang Gao ⓘ https://orcid.org/0009-0005-6735-5353
Hui Chen ⓘ https://orcid.org/0000-0002-7218-9047
Geyang Xu ⓘ https://orcid.org/0000-0002-5542-1350

### Ethics

Written informed consent was obtained from each donor. The study protocol was approved by the Institutional Review Board of Jinan University. Mucosal biopsies were obtained from human intestines by using a colonoscopy (CF HQ290I; Olympus).
The animal protocols were approved by the Animal Care and Use Committee of Jinan University (IACUC-20230517-01).

Reviewer #1 (Public review): https://doi.org/10.7554/eLife.97854.3.sa1
Reviewer #2 (Public review): https://doi.org/10.7554/eLife.97854.3.sa2
Reviewer #3 (Public review): https://doi.org/10.7554/eLife.97854.3.sa3
Author response https://doi.org/10.7554/eLife.97854.3.sa4

## Additional files

### Supplementary files
• MDAR checklist

### Data availability
All of the data supporting the findings of this study are included in the article and source data files.

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
