## [Editor Report · eLife Assessment]

This study focuses on the regulation of GLP-1 in enteroendocrine L cells and how this may be stimulated by the mechanogated ion channel Piezo1 and the CaMKKbeta-CaMKIV-mTORC1 signaling pathway. The work is innovative and is considered **valuable**, as the hypothesis that is being tested may have significant mechanistic and translational implications. Data to support the proposed mechanism were considered incomplete, yet data to support the overall physiological characterization were considered **solid**.

---

## [Referee Report · Reviewer #1 (Public review)]

Summary:

In this manuscript, authors intended to prove that gut GLP-1 expression and secretion can be regulated by Piezo1, and hence by mechanistic/stretching regulation. For this purpose, they have assessed Piezo1 expression in STC-1 cell line (a mouse GLP-1 producing cell line) and mouse gut, showing the correlation between Piezo1 level and Gcg levels (Fig. S1). They then aimed to generate gut L cell-specific Piezo1 KO mice and claimed the mice show impaired glucose tolerance and GLP-1 production, which can be mitigated by Ex-4 treatment (Fig. 1-2). Pharmacological agents (Yoda1 and GsMTx4) and mechanic activation (intestinal bead implantation) were then utilized to prove the existence of ileal Piezo1-regulated GLP-1 synthesis (Fig. 3). This was followed by testing such mechanism in a limited amount of primary L cells and mainly in the STC-1 cell line (Fig. 4-7).

While the novelty of the study is somehow appreciable, the bio-medical significance is not well demonstrated in the manuscript. The authors stated (in lines between lines 78-83) a number of potential side effects of GLP-1 analogs, how can the mechanistic study of GLP-1 production on its own be essential for the development of new drug targets for the treatment of diabetes. Furthermore, the study does not provide a clear mechanistic insight how the claimed CaMKKbeta/CaMKIV-mTORC1 signaling pathway upregulated both GLP-1 production and secretion. This reviewer also has concerns about the experimental design and data presented in the current manuscript, including the issue of how can proglucagon expression can be assessed by Western blotting.

Strengths:

Novelty of the concept.

Weaknesses:

Experimental design and key experiment information.

---

## [Referee Report · Reviewer #2 (Public review)]

Summary:

The study by Huang and colleagues focuses on GLP-1 producing enteroendocrine (EEC) L-cells and their regulation of GLP-1 production by a mechanogated ion channel Piezo1. The study describes Piezo1 expression by L-cells and using an exciting intersectional mouse model (villin to target epithelium and Gcg to target GLP-1 producing cells and others like glucagon producing pancreatic endocrine cells), which allows L-cell specific Piezo1 knockout. Using this model, they find an impairment of glucose tolerance, increased body weight, reduced GLP-1 content, and changes to the CaMKKbeta-CaMKIV-mTORC1 signaling pathway using normal diet and then high fat diet. Piezo1 chemical agonist and intestinal bead implantation reversed these changes and improved the disrupted phenotype. Using primary sorted L-cells and cell model STC-1, they found that stretch and Piezo1 activation increased GLP-1 and altered the molecular changes described above.

Strengths:

This is an interesting study testing a novel hypothesis that may have important mechanistic and translational implications. The authors generated an important intersectional genetics mouse model that allowed them to target Piezo1 L-cells specifically, and the surprising result of impaired metabolism is intriguing.

Weaknesses:

However, there are several critical limitations that require resolution before making the conclusions that the authors make. (1) A potential explanation for the data, and one that is consistent with existing literature [see for example, PMC5334365, PMC4593481], is that epithelial Piezo1, which is broadly expressed by the GI epithelium, impacts epithelial cell density and survival, and as such, if Piezo1 is involved in L-cell physiology, it may be through regulation of cell density. Thus, it is critical to determine L-cell densities and epithelial integrity in controls and Piezo1 knockouts systematically across the length of the gut, since the authors do not make it clear which gut region contributes to the phenotype they see. Current immunohistochemistry data are not convincing. (2) Calcium signaling in L-cells is implicated in their typical role of being gut chemosensors, and Piezo1 is a calcium channel, so it is not clear whether any calcium-related signaling mechanism would phenocopy these results. (3) Intestinal bead implantation, while intriguing, does not have clear mechanisms - and is likely to provide a point of intestinal obstruction and dysmotility. (4) previous studies, some that are very important, but not cited, contradict the presented results (e.g., epithelial Piezo1 role in insulin secretion) and require reconciliation.

Overall, this study makes an interesting observation but the data are not currently strong enough to support the conclusions.

- There needs to be data localizing Piezo1 to L-cells and importantly, this needs to be quantified - are all L-cells (small bowel and colon) Piezo1 positive? This is because several studies show Piezo1 affecting epithelial cell densities. If there are changes in L-cell or other EEC densities in Piezo1 knockout, that shift can potentially explain the changes that the authors see in glucose metabolism and weight.

- The intersectional model for L-cell transduction needs a deeper validation. Images in Fig 1e are not convincing for transduction of GFP in L-cells. The co-localization studies are not convincing, especially because Piezo1 labeling is very broad. There needs to be stronger validation of the intersectional Gcg-Villin-Piezo1 KO model. It is important to determine whether L-cell Piezo1 localization epithelium in small bowel and colon is present (above) and affected specifically in the knockout.

- The authors state that "Villin-1 (encoded by Vill1 gene) is expressed in the gastrointestinal epithelium, including L cells, but not in pancreatic α cells" (line 378-379). However, Villin is highly expressed in whole mouse islets (https://doi.org/10.1016/j.molmet.2016.05.015, Figure 1A).

- There needs to be quantification of L-cells in Piezo1 knockout. This is because several studies show Piezo1 affecting epithelial cell densities. If there are changes in L-cell or other EEC densities in Piezo1 knockout, that shift can potentially explain the changes that the authors see in glucose metabolism and weight.

- L-cells are classically considered to be chemosensors. Do nutritive signals, which presumably also increase calcium compete or complement or dominate L-cell GLP1 synthesis regulation?

- The mechanism of Glp1 synthesis vs release downstream of Piezo1 is not clear. The authors hypothesize that "Piezo1 might regulate GLP-1 synthesis through the CaMKKβ/CaMKIV-mTOR signaling pathway". However, references cited suggest that Ca2+ or cAMP lead to GLP-1-release, while mTOR primarily acts on the regulation of gene expression by promoting Gcg gene expression. These pathways do not clearly link to Piezo1 GLP-1 production. These mechanisms need to be reconciled.

- Previous study PMID 32640190 (not cited here) found that Villin-driven Piezo1 knockout, which knocks out Piezo1 from all epithelial intestinal cells (including L-cells), showed no significant alterations in blood glucose or body weight. This is opposite of the presented findings and therefore the current results require reconciliation.

Comments on revised version:

The authors have addressed several comments that were common to the reviewers - specificity and validity of the intersectional model, mechanism of signaling downstream of Piezo1 and reconciliation of the results with previous studies. The authors have provided extensive experiments and revisions which have made the manuscript stronger. However, many important questions remain, and unfortunately, the intersectional mouse model and mechanisms remain unclear.

- I appreciate the authors quantifying the density of L cells in the intersectional Piezo knockout. There is a very clear >50% drop-off in GLP-1+ cells with the Piezo1 knockout (Supp fig 7c, d). Interestingly, there was not a decrease in PYY+ cells, which is curious because GLP1 and PYY are co-expressed in L cells. The mechanism of regulation of one hormone but not the other in the same cell requires clarification and would be relevant for this work. To begin with, co-labeling PYY and GLP1 and showing that one hormone can be found without the other would be useful.

- Piezo1 immunofluorescence has very high background and overall poor specificity (Fig supp 5 and Fig supp 6B are good examples of poor Piezo1 immunofluorescence). Another method for labeling Piezo1 (e.g. via RNAscope) is required - and where tried (e.g., Fig 1L), the results are not convincing.

- The intersectional mouse model requires further validation. The data presented in Fig 1E do not help - the GFP positive cells do not look like L-cells and there appear to be GFP positive cells in the muscle and submucosa.

- Since Piezo1 is known to affect epithelial cell life span, barrier function maybe compromised. While I appreciate that the authors have obtain some images and measured zonular and occluded, this is unfortunately a suboptimal evaluation of barrier function.

- The mechanisms of calcium signaling that will presumably lead to GLP1 release due to Piezo1 activation and mTOR which authors link to GLP1 synthesis remain unreconciled.

- Intestinal bead implantation may provide an important area of obstruction, in addition to potential mechanical stimulation. Unfortunately whole gut transit time and fecal weight do not assay these functions well.

- I believe that the explanation regarding lack of previous findings connecting Piezo1 in the epithelium and glucose tolerance remain poorly reconciled with the current findings.

---

## [Referee Report · Reviewer #3 (Public review)]

Summary:

In this work, the authors proposed that the mechano-gated ion channel Piezo1 enhances GLP-1 production and secretion possibly through stimulating Ca2+-CaMKKbeta-CaMKIV-mTORC1 signaling pathway. By using intestinal L cell-specific piezo1 knock-out mice, intestinal bead implantation mice model, and the chemical agonist Yoda1, the authors claimed that piezo1 promotes pro-glucagon expression, GLP-1 production and secretion. In sorted primary intestinal L cells and STC-1 cells, the authors validated that CaMKKbeta-CaMKIV-mTORC1 signaling pathway positively regulated GLP-1 production and secretion. This study provides new evidence about the specific role of piezo1 in intestinal L cells, broadening the understanding of metabolic functions of piezo1.

Strengths:

The new concept and innovative in vivo and in vitro models.

Weaknesses:

Although the authors have addressed most of the issues in the revised manuscript, there are still some questions that need to be clarified.

(1) This study claimed that piezo1 enhances proglucagon expression, GLP-1 production and secretion through Ca2+-CaMKKbeta-CaMKIV-mTORC1 signaling pathway, which is a highly time-consuming process. However, as a mechano-gated ion channel, it should exert functions promptly. Is it possibly that piezo1 directly stimulates GLP-1 release by influx of Ca2+? if so, have authors measured intracellular Ca2+ concentration?

(2) The authors proposed that the CaMKKbeta-CaMKIV-mTORC1 signaling pathway mediated the effects of piezo1. However, the data is not convincing. At least, chemical inhibitors of CaMKKbeta/CaMKIV/mTORC1 should be used in intL-piezo1 KO mice or STC-1 cells to see if piezo1-induced GLP-1 secretion was abrogated by these chemical inhibitors.

(3) According to previous studies of the team, piezo1 could enhance insulin, ghrelin and GLP-1 secretion while inhibit glucagon production in pancreatic α-cells. In a recent work, the authors found that piezo1 in enterocytes suppresses nutrient absorption. Why an ion channel has these various effects in different cells? What is the fundamental and common mechanism underlying its metabolic functions? Its value as a drug target? These questions need to be discussed in more details.

---

## [Author Response]

The following is the authors’ response to the original reviews.

**Public Reviews:**

**Reviewer #1 (Public Review):**
Summary:In this manuscript, the authors intended to prove that gut GLP-1 expression and secretion can be regulated by Piezo1, and hence by mechanistic/stretching regulation. For this purpose, they have assessed Piezo1 expression in STC-1 cell line (a mouse GLP-1 producing cell line) and mouse gut, showing the correlation between Piezo1 level and Gcg levels (Figure S1). They then aimed to generate gut L cell-specific Piezo1 KO mice, and claimed the mice show impaired glucose tolerance and GLP-1 production, which can be mitigated by Ex-4 treatment (Figures 1-2). Pharmacological agents (Yoda1 and GsMTx4) and mechanic activation (intestinal bead implantation) were then utilized to prove the existence of ileal Piezo1-regulated GLP-1 synthesis (Figure 3). This was followed by testing such mechanism in a limited amount of primary L cells and mainly in the STC-1 cell line (Figures 4-7).While the novelty of the study is somehow appreciable, the bio-medical significance is not well demonstrated in the manuscript. The authors stated (in lines between lines 78-83) a number of potential side effects of GLP-1 analogs, how can the mechanistic study of GLP-1 production on its own be essential for the development of new drug targets for the treatment of diabetes. Furthermore, the study does not provide a clear mechanistic insight on how the claimed CaMKKbeta/CaMKIV-mTORC1 signaling pathway upregulated both GLP-1 production and secretion. This reviewer also has concerns about the experimental design and data presented in the current manuscript, including the issue of how proglucagon expression can be assessed by Western blotting.Strengths:The novelty of the concept.Weaknesses:Experimental design and key experiment information.

We appreciate the reviewer's comments. Nowadays, GLP-1-based therapy is well-recognized and commonly used in treatment of Type 2 Diabetes Mellitus (T2DM). Therefore, elucidation of the mechanism that regulates GLP-1 production is essential for the development of new drug targets for the treatment of diabetes. We have revised the relevant wording in the manuscript.

In our previous studies, we have elucidated the role of mTOR/S6K pathway in regulating GLP-1 production in L cells. Using STC-1 cell line and different mouse models, including *Neurog3-Tsc1−/−* mice, rapamycin or L-lucine treatment to stimulate mTOR activity, we have demonstrated that mTOR stimulates proglucagon gene expression and thus GLP-1 production (Diabetologia 2015;58(8):1887-97； Mol Cell Endocrinol. 2015 Nov 15:416:9-18.). Based on our previous studies, we found that Piezo1 regulated mTOR/S6K pathway and thus proglucagon expression and GLP-1 production through a Ca2+/CaMKKbeta/CaMKIV pathway in our present study. Although we could not exclude involvement of other signaling pathways downstream of Piezo1 in regulating the cleavage of proglucagon, granule maturation and the final release of GLP-1, our present study provided evidence to support the involvement of the Ca2+/CaMKKbeta/CaMKIV/mTOR pathway in mediating the role Piezo1 in proglucagon expression and GLP-1 production.

The reviewer also expressed concerns on the use of western blot to detect proglucagon expression. Proglucagon is encoded by the *GCG* gene and is cleaved by PC1/3 in L cells to form mature GLP-1. In fact, measurement of intestinal proglucagon protein is a common approach for assessing GLP-1 production in the intestine. Here are some examples from other researchers: Diabetes. 2013 Mar;62(3):789-800. Gastroenterology. 2011 May;140(5):1564-74. 2004 Jul 23;279(30):31068-75. The proglucagon antibody used in our study was purchased from abcam (Cat#ab23468), which can detect proglucagon at 21 kDa.

**Reviewer #2 (Public Review):**
Summary:The study by Huang and colleagues focuses on GLP-1 producing entero-endocrine (EEC) L-cells and their regulation of GLP-1 production by a mechano-gated ion channel Piezo1. The study describes Piezo1 expression by L-cells and uses an exciting intersectional mouse model (villin to target epithelium and Gcg to target GLP-1-producing cells and others like glucagon-producing pancreatic endocrine cells), which allows L-cell specific Piezo1 knockout. Using this model, they find an impairment of glucose tolerance, increased body weight, reduced GLP-1 content, and changes to the CaMKKbeta-CaMKIV-mTORC1 signaling pathway using a normal diet and then high-fat diet. Piezo1 chemical agonist and intestinal bead implantation reversed these changes and improved the disrupted phenotype. Using primary sorted L-cells and cell model STC-1, they found that stretch and Piezo1 activation increased GLP-1 and altered the molecular changes described above.Strengths:This is an interesting study testing a novel hypothesis that may have important mechanistic and translational implications. The authors generated an important intersectional genetics mouse model that allowed them to target Piezo1 L-cells specifically, and the surprising result of impaired metabolism is intriguing.Weaknesses:However, there are several critical limitations that require resolution before making the conclusions that the authors make.(1) A potential explanation for the data, and one that is consistent with existing literature [see for example, PMC5334365, PMC4593481], is that epithelial Piezo1, which is broadly expressed by the GI epithelium, impacts epithelial cell density and survival, and as such, if Piezo1 is involved in L-cell physiology, it may be through regulation of cell density. Thus, it is critical to determine L-cell densities and epithelial integrity in controls and Piezo1 knockouts systematically across the length of the gut, since the authors do not make it clear which gut region contributes to the phenotype they see. Current immunohistochemistry data are not convincing.

We appreciate the reviewer's comment and agree that Piezo1 may impact L-cell density and epithelial integrity. To address this, we have incorporated quantification of L-cell density in new Figure Supplement 7. The quantitative results demonstrate that the specific deletion of the piezo1 gene in L cells did not significantly impact L-cell density.

Regarding epithelial integrity, we assessed the expression of tight junction proteins (ZO-1 and Occludin). As demonstrated in new Figure Supplement 8, the expression of tight junction proteins such as ZO-1 and Occludin did not show significant changes in *IntL-Piezo1-/-* mice compared to littermate controls.

Furthermore, we conducted double immunofluorescence of Piezo1 and GLP-1 in the duodenum, jejunum, ileum, and colon of control and *IntL-Piezo1-/-* mice. As illustrated in new Figure Supplement 5, Piezo1 is expressed in GLP-1-positive cells of the duodenum, jejunum, ileum, and colon of control mice, but not in *IntL-Piezo1-/-* mice.

(2) Calcium signaling in L-cells is implicated in their typical role of being gut chemo-sensors, and Piezo1 is a calcium channel, so it is not clear whether any calcium-related signaling mechanism would phenocopy these results.

We agree with the reviewer that Piezo1 is a calcium channel (validation of the Ca2+ influx-mediated Piezo1 in primary L cells and STC-1 cells are shown in figure 4A-C and figure 5A-C). According to our study, calcium-related signaling mechanism such as calcium/calmodulin-dependent protein kinase kinase 2 (CaMKKβ) -Calcium/Calmodulin Dependent Protein Kinase IV (CaMKIV) may contribute the phenotype seen in the _IntL-Piezo1-/_mice. In addition, we also discussed other potential calcium-related signaling mechanisms in the article's discussion section (lines645-656).

(3) Intestinal bead implantation, while intriguing, does not have clear mechanisms and is likely to provide a point of intestinal obstruction and dysmotility.

We appreciate the reviewer’s comment. To ascertain if intestinal bead implantation led to intestinal obstruction and dysmotility, we conducted a bowel transit time test and detected the postoperative defecation (As shown in new Figure Supplement 9). The results revealed no difference in bowel transit time and fecal mass between the sham-operated mice and those implanted with beads. Furthermore, to assess whether the animals were in pain or under any discomfort after intestinal bead implantation, we performed abdominal mechanical sensitivity test three days after the surgery. As indicated in Figure Supplement 9C, no difference in abdominal pain threshold was observed between sham and bead-implanted mice. These results suggest that the mice did not experience discomfort during the experiment.

(4) Previous studies, some that are very important, but not cited, contradict the presented results (e.g., epithelial Piezo1 role in insulin secretion) and require reconciliation.

Thanks a lot for the point. We have cited more previous studies. The lack of changes in blood glucose seen in *Villin-Piezo1-/-* mice reported by Sugisawa et. al. is not surprising (Cell. 2020 Aug 6;182(3):609-624.e21.). Actually, in another recent study from our group, we found similar results when the *Villin-Piezo1-/-* mice *Piezo1fl/fl* control mice were fed with normal chow diet. Since Villin-1 is expressed in all the epithelial cells of the gut, including enterocytes and various types of endocrine cells, the effect of L-cell Piezo1 loss may be masked by other cell types under normal condition. However, impaired glucose tolerance was seen in *Villin-Piezo1-/-* mice compared to the *Piezo1fl/fl* control mice after high fat diet for 8 weeks. We further found that Piezo1 in enterocytes exerted a negative effect on the glucose and lipid absorption. Loss of Piezo1 in enterocytes led to over-absorption of nutrients under high-fat diet. （Tian Tao, Qing Shu, Yawen Zhao, Wenying Guo, Jinting Wang, Yuhao Shi, Shiqi Jia, Hening Zhai, Hui Chen, Cunchuan Wang*, Geyang Xu*, Mechanical regulation of lipid and sugar absorption by Piezo1 in enterocytes, Acta Pharmaceutica Sinica B , Accepted, 2024. (https://doi.org/10.1016/j.apsb.2024.04.016).

Overall, this study makes an interesting observation but the data are not currently strong enough to support the conclusions.
**Recommendations for the authors:**

**Reviewer #1 (Recommendations For The Authors):**
Major concerns(1) Figure 1L was labeled wrong, and the co-localization was not clear. The KO leads to such a strong effect on the percentage of GLP-1 positive cells (panel M) but was not clearly demonstrated with immune-staining. Additional experiments are needed to prove tissue-specific knockout in gut GLP-1-producing cells only, but not in other cell lineages or elsewhere. If so, how was the change in gut Gcg mRNA expression? Importantly, this review is not clear on how to use Western blotting to measure proglucagon expression in the tissue samples. What is the size of the product? The antibody information was not provided in the manuscript. Figure 1N, a potential mechanism that affects GLP-1 production involving mTORC and downstream molecules. This comes from nowhere.

We appreciate the reviewer's feedback. The incorrect label has been corrected in the new Figure 1L. As suggested, we have performed additional experiments to demonstrate tissue-specific knockout of Piezo1 in gut GLP-1-producing cells exclusively, excluding other cell lineages or locations.

As shown in Figure Supplement 6, Piezo1 remains expressed in ileal ghrelin-positive cells and pancreatic glucagon-positive cells of IntL-Piezo1-/mice, suggesting that Piezo1 was specifically knocked out in L cells, but not in other endocrine cell types. Furthermore, the decrease was only observed in GLP-1 levels, but not PYY levels, in L cells of *IntL-Piezo1-/-* mice compared to controls, suggesting that the loss of Piezo1 in L cells affects GLP-1 levels specifically, but not the secretion of other hormones produced by L cells (Figure Supplement 7A-D).

In our previous studies, we have elucidated the role of mTOR/S6K pathway in regulating GLP-1 production in L cells. Using STC-1 cell line and different mouse models, including *Neurog3-Tsc1−/−* mice, rapamycin or L-lucine treatment to stimulate mTOR activity, we have demonstrated that mTOR stimulates proglucagon gene expression and thus GLP-1 production (Diabetologia 2015;58(8):1887-97； Mol Cell Endocrinol. 2015 Nov 15:416:9-18.). Based on our previous studies, we found that Piezo1 regulated mTOR/S6K pathway and thus proglucagon expression and GLP-1 production through a Ca2+/CaMKKbeta/CaMKIV pathway in our present study.

Although we could not exclude involvement of other signaling pathways downstream of Piezo1 in regulating the cleavage of proglucagon, granule maturation and the final release of GLP-1, our present study provided evidence to support the involvement of the Ca2+/CaMKKbeta/CaMKIV/mTOR pathway in mediating the role Piezo1 in proglucagon expression and GLP-1 production.

The reviewer also expressed concerns on the use of western blot to detect proglucagon expression. Proglucagon is encoded by the *GCG* gene and is cleaved by PC1/3 in L cells to form mature GLP-1. In fact, measurement of intestinal proglucagon protein is a common approach for assessing GLP-1 production in the intestine. Here are some examples from other researchers: Diabetes. 2013 Mar;62(3):789-800. Gastroenterology. 2011 May;140(5):1564-74. 2004 Jul 23;279(30):31068-75. The proglucagon antibody used in our study was purchased from abcam (Cat#ab23468), which can detect proglucagon at 21 kDa.

(2) In Figure 2, the LFD control mouse group was missing. Again, I don't understand the detection of proglucagon by Western blotting in this figure.

We appreciate the reviewer's comments. The figure 1 presents the phenotypic changes of transgenic mice under low-fat diet feeding, while figure 2 focuses on the phenotypic changes of transgenic mice under high-fat diet feeding. As we mentioned before, western blot is often used in detection of the precursor of GLP-1 named proglucagon.

(3) Why show body weight change but not body weight itself? How are the changes compared (which one serves as the control)? Again, how to do Western blotting on pro-glucagon detection?

We appreciate the reviewer's comments. Body weight has been added in new figure3. Proglucagon is the precursor of GLP-1. Intestinal proglucagon protein measurement is commonly used to assess GLP-1 production in the intestine.

(4) After reading the whole manuscript, this reviewer cannot get a clear picture of how the claimed CaMKKbeta-mTORC1 pathway mediates the function of Pieo1 activation (via the utilization of Yoda1 or intestinal bead implantation) on Gcg expression (at the transcription level or mRNA stability level?), hormone production, the genesis of GLP-1 producing cells, and the secretion of the hormone.

We appreciate the reviewer's comments. Figure 7 showed that overexpression of CaMKKbeta and CaMKIV enhanced mTOR and S6K phosphorylation, proglucagon expression and GLP-1 secretoin, while CaMKKbeta inhibitor STO609 inhibited mTOR and S6K phosphorylation, proglucagon expression and GLP-1 secretoin, suggesting CaMKKbeta and CaMKIV was involved in GLP-1 production. Moreover, mTOR inhibitor rapamycin inhibited Yoda1-induced proglucagon expression and GLP-1 secretion. These results suggested that CaMKKbeta/CaMKIV/mTOR mediated the effect of Piezo1 on GLP-1 production.

I strongly suggest that authors focus on more solid findings and dissect the mechanistic insight on something more meaningful, but not on everything (hormone coding gene expression, hormone production, and hormone secretion).

GLP-1 production involves multiple steps, including proglucagon expression, protein cleavage, granule packaging and final release. In our present study, we focused on how mechanical signals regulated proglucagon expression in L-cells and thus promote GLP-1 production. We did not exclude the possibility that mechanical signals could also affect other step of GLP-1 production and we discussed this possibility in the discussion section.

Minor concerns(1) Figure S1A. STC-1 is a Gcg expression cell line, which shows less amount of Peio1 mRNA when compared with most primary tissue samples tested. This does not support the fundamental role of Peio1 in regulating Gcg expression. Maybe qRT-PCR will be more helpful for establishing the correlation.

Thanks a lot for the comments. As suggested, the results of qRT-PCR have been added in new Figure S1A.

(2) There are numerous scientific presentation problems in the written manuscript. Necessary literature citations are missing especially for key methods (such as bean implantation).

Thank you very much for your comments. We have made every effort to enhance the scientific presentation and have included the necessary literature citations.

**Reviewer #2 (Recommendations For The Authors):**
Overall, this study makes an interesting observation but the data are not currently strong enough to support the conclusions.(1) There needs to be data localizing Piezo1 to L-cells and importantly, this needs to be quantified - are all L-cells (small bowel and colon) Piezo1 positive?

Thank you very much for your comments. We performed double immunofluorescence of Piezo1 and GLP-1 in the duodenum, jejunum, ileum, and colon of control and *IntL-Piezo1-/-* mice. As shown in new Figure Supplement 5, Piezo1 is expressed in about 90% of GLP-1-positive cells in the duodenum, jejunum, ileum, and colon of control mice, but not in *IntL-Piezo1-/-* mice.

(2) The intersectional model for L-cell transduction needs deeper validation. Images in Figure 1e are not convincing for the transduction of GFP in L-cells. The co-localization studies are not convincing, especially because Piezo1 labeling is very broad. There needs to be stronger validation of the intersectional Gcg-Villin-Piezo1 KO model. It is important to determine whether L-cell Piezo1 localization epithelium in the small bowel and colon is present (above) and affected specifically in the knockout.

Thanks a lot for the comments. In our study, we conducted a double immunofluorescence analysis for Piezo1 and GLP-1 across various segments of the gastrointestinal tract, including the duodenum, jejunum, ileum, and colon, in both control and *IntL-Piezo1-/-* mice. As illustrated in the newly incorporated Figure Supplement 5, it was observed that Piezo1 is indeed expressed within the cells of the aforementioned gastrointestinal segments in control mice, which are also positive for GLP-1 expression. In stark contrast, no evidence of Piezo1 expression was detected in the *IntL-Piezo1-/-* mice. Consistent with these findings, in situ hybridization experiments corroborated the absence of Piezo1 expression within GLP-1 positive cells in the *IntL-Piezo1-/-* mice, offering evidence for the successful knockout of Piezo1 in the L cells of these knockout mice. (Figure 1L and M).

In Figure 1E, IntL-Cre mice were bred with mT/mG reporter mice to further validate Cre recombinase activity and specificity. All tissues and cells of mT/mG mice express red fluorescence (membrane-targeted tdTomato; mT) at baseline, and switch to membrane-targeted EGFP in the presence of cell-specific Cre. EGFP expression was only observed scatteredly in the intestine, but not in the pancreas, indicating the intestinal-specific Cre activity in the IntL-Cre mice (Figure 1E). We have revised the relevant expressions in the main text.

(3) The authors state that "Villin-1 (encoded by Vill1 gene) is expressed in the gastrointestinal epithelium, including L cells, but not in pancreatic α cells" (lines 378-379). However, Villin is highly expressed in whole mouse islets (https://doi.org/10.1016/j.molmet.2016.05.015, Figure 1A).

Thanks a lot for the comments. Although Hassan Mziaut et al. reported that Villin is highly expressed in whole mouse islets, in that article, only the co-localization of insulin cells with Villin is mentioned, while the co-localization of glucagon and Villin is lacking.

According to our research (refer to Author response image 1 below) and previous study (Rutlin, M. et al, 2020, The Villin1 Gene Promoter Drives Cre Recombinase Expression in Extraintestinal Tissues. Cell Mol Gastroenterol Hepatol, 10(4), 864-867.e865.), Villin is sparsely expressed in pancreatic tissue but not highly expressed in islets. We did not observed co-localization of glucagon and Villin in the pancreas (see Author response image 1A and B below). The same antibody was used to stain intestine, which show specific expression on the apical side of the intestinal villi (see Author response image 1C below).

**Author response image 1. sa4fig1:** 

(4) There needs to be quantification of L-cells in Piezo1 knockout. This is because several studies show Piezo1 affecting epithelial cell densities. If there are changes in L-cell or other EEC densities in Piezo1 knockout, that shift can potentially explain the changes that the authors see in glucose metabolism and weight.

We appreciate the reviewer’s comment. We agree that Piezo1 may affect L-cell density and epithelial integrity.

To assess epithelial integrity we examined the expression of tight junction proteins (ZO-1 and Occludin). As shown in new Figure Supplement 8, the expression of tight junction proteins, including ZO-1 and Occludin, remained unchanged in *IntL-Piezo1-/-* mice when compared to littermate controls.

To assess the L-cell density, we stained PYY, another hormone mainly secreted by L cells, in both control and *IntL-Piezo1-/-* mice. As shown in new Figure Supplement 7A and B, the percentage of PYY positive cells were not significantly different between control and *IntL-Piezo1-/-* mice, suggesting that the L-cell density was not affected by Piezo1 knockout.

(5) L-cells are classically considered to be chemosensors. Do nutritive signals, which presumably also increase calcium compete or complement or dominate L-cell GLP1 synthesis regulation?

We appreciate the reviewer ’ s comment and agree that L-cells are traditionally considered to be chemosensors. It is also recognized that nutritive signals regulate L-cell GLP1 synthesis. We have addressed these points in lines 568-595. Both nutritive and mechanical signals regulate GLP-1 production. While the food needs to be digested and nutrients absorbed before L-cells can detect the nutritive signals, mechanical stimulation provides a more direct and rapid response. However, determining whether nutritive signals compete, complement with mechanical signals or dominate in L-cell GLP-1 production will require to be further explored.

(6) The mechanism of Glp1 synthesis vs release downstream of Piezo1 is not clear. The authors hypothesize that "Piezo1 might regulate GLP-1 synthesis through the CaMKKβ/CaMKIV-mTOR signaling pathway". However, references cited suggest that Ca2+ or cAMP leads to GLP-1-release, while mTOR primarily acts on the regulation of gene expression by promoting Gcg gene expression. These pathways do not clearly link to Piezo1 GLP-1 production. These mechanisms need to be reconciled.

Thanks a lot for the point. The effect of Piezo1-mediated Ca2+ increase on GLP-1 production may be two-fold: promote Gcg gene expression through CaMKKβ/CaMKIV-mTOR and promote GLP-1 release by degranulation. Both gene expression and release are important to sustained GLP-1 production.

(7) Previous study PMID 32640190 (not cited here) found that Villin-driven Piezo1 knockout, which knocks out Piezo1 from all epithelial intestinal cells (including L-cells), showed no significant alterations in blood glucose or body weight. This is the opposite of the presented findings and therefore the current results require reconciliation.

We have cited PMID 32640190 in our revised manuscript. The lack of changes in blood glucose seen in *Villin-Piezo1-/-* mice reported by Sugisawa et. al. is not surprising (Cell. 2020 Aug 6;182(3):609-624.e21.). Actually, in another recent study from our group, we found similar results when the _Villin-Piezo1-/_mice *Piezo1fl/fl* control mice were fed with normal chow diet. Since Villin-1 is expressed in all the epithelial cells of the gut, including enterocytes and various types of endocrine cells, the effect of L-cell Piezo1 loss may be masked by other cell types under normal condition. However, impaired glucose tolerance was seen in *Villin-Piezo1-/-* mice compared to the *Piezo1fl/fl* control mice after high fat diet for 8 weeks. We further found that Piezo1 in enterocytes exerted a negative effect on the glucose and lipid absorption. Loss of Piezo1 in enterocytes led to over-absorption of nutrients under high-fat diet (Tian Tao, Qing Shu, Yawen Zhao, Wenying Guo, Jinting Wang, Yuhao Shi, Shiqi Jia, Hening Zhai, Hui Chen, Cunchuan Wang, Geyang Xu, Mechanical regulation of lipid and sugar absorption by Piezo1 in enterocytes, Acta Pharmaceutica Sinica B, Accepted, 2024, https://doi.org/10.1016/j.apsb.2024.04.016).

**Reviewing Editor (Recommendations For The Authors):**
Your paper - while innovative in concept and interesting - has many flaws that in my opinion need to be corrected before the paper and pre-print should be published or uploaded as pre-print. Can you please make every effort to address the missing data that the Reviewers have asked for and correct the lack of references as noted in the reviews? Thank you.

Thank you for the invaluable suggestions provided by the editors and reviewers. In response to these suggestions, we have included the missing data as requested and rectified the lack of references to the best of our ability. We hope that these revisions will effectively address the concerns raised by the editors and reviewers.